# CIRCUIT TRANSFORMER: A TRANSFORMER THAT PRESERVES LOGICAL EQUIVALENCE

**Xihan Li[1], Xing Li[2], Lei Chen[2], Xing Zhang[2], Mingxuan Yuan[2], Jun Wang[1]**
[1] UCL Centre for Artificial Intelligence     [2] Huawei Noah's Ark Lab
{xihan.li,jun.wang}@cs.ucl.ac.uk
{li.xing2,lc.leichen,zhangxing85,Yuan.Mingxuan}@huawei.com

## ABSTRACT

Implementing Boolean functions with circuits consisting of logic gates is fundamental in digital computer design. However, the implemented circuit must be exactly equivalent, which hinders generative neural approaches on this task due to their occasionally wrong predictions. In this study, we introduce a generative neural model, the "Circuit Transformer", which eliminates such wrong predictions and produces logic circuits strictly equivalent to given Boolean functions. The main idea is a carefully designed decoding mechanism that builds a circuit step-by-step by generating tokens, which has beneficial "cutoff properties" that block a candidate token once it invalidate equivalence. In such a way, the proposed model works similar to typical LLMs while logical equivalence is strictly preserved. A Markov decision process formulation is also proposed for optimizing certain objectives of circuits. Experimentally, we trained an 88-million-parameter Circuit Transformer to generate equivalent yet more compact forms of input circuits, outperforming existing neural approaches on both synthetic and real world benchmarks, without any violation of equivalence constraints.

Code: https://github.com/snowkylin/circuit-transformer

## 1 INTRODUCTION

In this work, we are concerned about the feasibility of generative neural models on a foundational logic problem — circuit realization. Given a Boolean function, it is required to produce a directed acyclic graph (DAG) that connects basic logic gates (AND, OR, NOT, etc.), whose output exactly matches the given function. The DAG is also named a *circuit*[1]. Circuit realization is not only of theoretical significance in circuit complexity research (Shannon, 1949; Vollmer, 1999; Sipser, 2013), but also lies in the core of digital design (Wang et al., 2009).

Recently, generative neural models, represented by large language models (LLMs) that recurrently predict next token, excel in multiple domains from natural conversation to code generation (Pei et al., 2024; Liu et al., 2023a). However, their feasibility for circuit realization is questionable. The main barrier is the requirement of strictly preserving *logical equivalence*. Given a Boolean function of $N$ inputs, there are $2^N$ possible combinations of input values, and the output of the realized circuit must be equal to the given function on exactly all the $2^N$ values. An example is shown in Figure 1. For generative neural models, due to their predictive and data-driven nature, they occasionally make wrong predictions and fall short in maintaining complex logical relations, which lead researchers to believe that they are less promising in such equivalence-preserving circuit realization (Huang et al., 2021). Instead, recent machine learning approaches in this topic mainly focus on strengthening traditional symbolic methods, replacing certain modules to a learned one to improve the performance.

In this work, we show that a formal guarantee of logical equivalence is achievable for generative neural models with next token prediction. A novel mechanism is developed to build a circuit incrementally by predicting a sequence of tokens, which is carefully designed to incorporate beneficial

---

[1]More specifically, a combinatorial logic circuit, which is referred to as "circuit" in short in this paper unless otherwise stated.

| $x_3$ | $x_2$ | $x_1$ | $x_0$ | $y_1$ | $y_0$ | $x_3$ | $x_2$ | $x_1$ | $x_0$ | $y_1$ | $y_0$ |
|---|---|---|---|---|---|---|---|---|---|---|---|
| 0 | 0 | 0 | 0 | 0 | 1 | 1 | 0 | 0 | 0 | 0 | 1 |
| 0 | 0 | 0 | 1 | 0 | 0 | 1 | 0 | 0 | 1 | 0 | 0 |
| 0 | 0 | 1 | 0 | 0 | 0 | 1 | 0 | 1 | 0 | 0 | 0 |
| 0 | 0 | 1 | 1 | 1 | 1 | 1 | 0 | 1 | 1 | 1 | 1 |
| 0 | 1 | 0 | 0 | 0 | 1 | 1 | 1 | 0 | 0 | 0 | 0 |
| 0 | 1 | 0 | 1 | 0 | 0 | 1 | 1 | 0 | 1 | 1 | 1 |
| 0 | 1 | 1 | 0 | 0 | 0 | 1 | 1 | 1 | 0 | 0 | 0 |
| 0 | 1 | 1 | 1 | 1 | 1 | 1 | 1 | 1 | 1 | 1 | 1 |

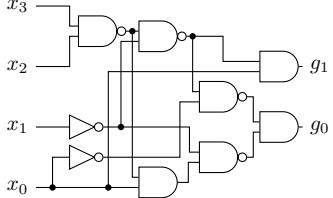
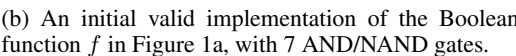

(a) A Boolean function $(y_1, y_0) = f(x_3, x_2, x_1, x_0)$, in which $x_0, x_1, x_2, x_3, y_0, y_1 \in \{0, 1\}$.

(b) An initial valid implementation of the Boolean function $f$ in Figure 1a, with 7 AND/NAND gates.

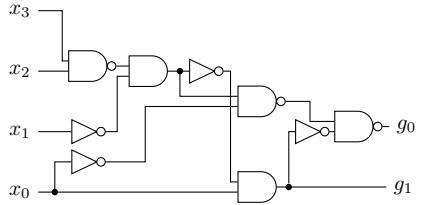
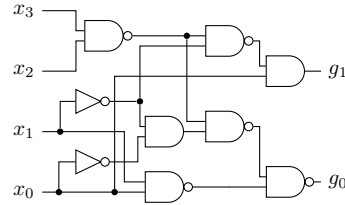

(c) A valid implementation of $f$ with 5 AND/NAND gates, which is optimal in size.

(d) An invalid implementation. In this circuit $g_0 = 0 \neq y_0$ when $\boldsymbol{x} = (1, 1, 0, 1)$.

Figure 1: An example showing how a Boolean function can be implemented by circuits, i.e., cascade connections of logic gates. $\square$ and $\triangleright\!\circ$ are the AND gate and NOT gate respectively, and $\square\!\!\circ$ is the NAND gate, an AND gate followed by a NOT gate. The circuits shown in Figure 1b and Figure 1c are both valid implementations of $f$, while the latter is more compact in size. The circuit shown in Figure 1d is invalid, even if the difference is as small as one bit.

"cutoff properties". Given a partially-defined circuit and a candidate token, it is possible to quickly determine whether adding this candidate token will violate the logical equivalence. In such a way, when predicting a new token, candidates that invalidate equivalence will be blocked, ensuring the logical equivalence to be preserved throughout the generation process. The mechanism is also designed to eliminate "dead ends", which means adding a valid new token is always possible until the circuit is completed. Therefore, our proposed mechanism allows smooth generation of circuits without backtracking, analogous to typical LLM-based natural language generation.

Based on the above mechanism, we propose the Circuit Transformer, which adopts the classical encoder-decoder Transformer architecture (Vaswani et al., 2017). The Boolean function to be implemented is embedded by the Transformer encoder, and the aforementioned decoding mechanism is adopted by inserting a masking layer at the end of the Transformer decoder. When predicting the next token $s_{t+1}$, the masking layer blocks candidates that invalidate equivalence, based on the current "cutoff" circuit partially defined by $s_1, \ldots, s_t$. Such a Transformer can be trained in a standard supervised way, while the equivalence of produced circuits is guaranteed during the inference stage.

Moreover, our proposed approach enables optimization methods to search freely within the equivalence class to optimize a certain objective. Specifically, equivalent circuit generation with a certain objective can be formulated as a Markov decision process, and an example is provided to minimize circuit size by designating a proper reward function.

To demonstrate the effectiveness of our proposed techniques, a Circuit Transformer of 88 million parameters is trained in a supervised way to generate strictly equivalent yet more compact implementations of given circuits. Experimental results show that the trained model is capable of generating *strictly equivalent* implementations for all unseen circuits in the test set, and the size decrease is close to the traditional method that serves as the supervised signal.

To conclude, we make the following main contributions:

- A decoding mechanism that builds a circuit by a sequence of tokens, with beneficial cutoff properties that allows logical equivalence to be preserved throughout the decoding process.
- A generative neural model named "Circuit Transformer" adopting the proposed decoding mechanism as a masking layer, which can be trained normally and preserve equivalence during the inference stage.
- A formulation of equivalence-preserving circuit optimization as a Markov decision process.
- Extensive experiments on the circuit size minimization problem demonstrating the equivalence-preserving capability of Circuit Transformer.

## 2 RELATED WORK

While data-driven AI techniques achieve tremendous success in recent years, they are generally based on probabilistic prediction that allows occasional mistakes, thus less promising to be directly applied to domains requiring exact preciseness, such as theorem proving and circuit design. Therefore, the mainstream paradigm of AI for such domains is to aid traditional methods in solving *sub-problems* that are more relaxed with respect to exactness. For circuit design, such relaxed sub-problems include SAT solver acceleration (Selsam & Bjørner, 2019; Wang et al., 2023; Zhang et al., 2020; Guo et al., 2023), circuit representation learning (Zhang et al., 2019; Yang et al., 2022b; Li et al., 2022; Wang et al., 2022), learning based graph optimization (Neto et al., 2021; Li et al., 2023), operator sequence scheduling (Yu et al., 2018; Hosny et al., 2020; Grosnit et al., 2022; Zhu et al., 2020; Yang et al., 2022a), and placement and routing (Mirhoseini et al., 2021; Cheng et al., 2022).

Nonetheless, there are a few research works (Schmitt et al., 2021; 2023; d'Ascoli et al., 2023) that attempt to generate circuits or logical expressions directly via next token prediction models. However, feasibility guarantee is not considered in these work, resulting in different extent of constraint violation in their reported result. In (Schmitt et al., 2021), 20% - 70% generated circuits violate the input specification in different datasets, while a follow-up work (Schmitt et al., 2023) mitigates the invalid percentage to 16% - 65% with pre-trained language models. In (d'Ascoli et al., 2023), 5% - 10% cases failed to be fully recovered when the number of operators are between 25 and 50. Apart from the low-level circuit generation, many researchers focus on transferring the software code generation capability of LLM to high-level hardware code generation, but these methods still suffer from functional inequivalence (Liu et al., 2023b; Thakur et al., 2024; Pei et al., 2024; Liu et al., 2024).

There are also research works that guarantee the feasibility of solutions via action masking, especially for problems involving routing such as maze game, traveling salesman problem, and vehicle routing problem (Nazari et al., 2018; Kool et al., 2019; Duan et al., 2020). However, these problems are usually intuitive to be sequentially modelled, allowing action masks to simply filter invalid actions such as wall-hitting directions and already visited cities. Action masking techniques under complicated constraints have yet to be explored.

For sequential representation of gate-level circuits, existing formats include graph-based ones like AIGER (Biere, 2007) and BLIF (Berkeley, 1992), and code-based ones like Verilog (Thomas & Moorby, 2008) and VHDL (Skahill, 1996). However, they can only be validated once the content is given in full, lacking the "cutoff properties" that will be discussed in Section 4.

## 3 PROBLEM DESCRIPTION

In this work, we focus on formally guaranteeing a generative neural model to produce valid circuits that strictly match a given Boolean function. For an $N$-input, $M$-output Boolean function

$$\boldsymbol{y} = f(\boldsymbol{x}), \tag{1}$$

in which $\boldsymbol{x} = (x_1, \ldots, x_N) \in \{0,1\}^N$ and $\boldsymbol{y} = (y_1, \ldots, y_M) \in \{0,1\}^M$, any valid circuit $g$ implementing the function must be strictly within the equivalence class of $f$. That is, for any $N$-dimensional input $\boldsymbol{x} \in \{0,1\}^N$ (there are $2^N$ of them), the $M$-dimensional output of the circuit, denoted as $g(\boldsymbol{x})$, must be equal to $f(\boldsymbol{x})$ in an element-wise way. This can be formally described as $g \in C(f)$, in which

$$C(f) = \{g \mid g(\boldsymbol{x}) = f(\boldsymbol{x}), \forall \boldsymbol{x} \in \{0,1\}^N\} \tag{2}$$

is the equivalence class of $f$. Any violation of the $2^N$ equivalence constraints will disqualify the circuit as a valid implementation of $f$. An example with a 4-input, 2-output Boolean function is shown in Figure 1, in which the produced circuits must satisfy $2^4 = 16$ equality constraints.

## 4 METHODS

In this section, we start from a general introduction of constrained sequence generation, highlighting the importance of "cutoff properties" for efficiency. Then we propose a sequential representation of

circuits in Section 4.2 that has such beneficial properties, allowing efficient generation akin to natural language generation while strictly adhering to equivalence constraints. It is followed by a postprocessing step regarding graph structures in Section 4.3. Given the circuit representation, we show in Section 4.4 how it can be integrated into the encoding and decoding process of a Transformer model, forming the Circuit Transformer. Finally, we discuss in Section 4.5 how our proposed approach help model equivalence-preserving circuit optimization as a Markov decision process.

## 4.1 Constrained Sequence Generation with Cutoff Properties

In this section, we briefly introduce how to generate sequences $s_1, \ldots, s_n$, $s_i \in D$ with constraints, i.e., certain property $F(s_1, \ldots, s_n) \in \{0, 1\}$ strictly holds. We use the eight-queen puzzle as an illustrative example, while circuit-specific sequence generation will be followed in the next section. In this puzzle, we place eight chess queens on an $8 \times 8$ chessboard. The queen positions can be encoded as a sequence $s_1, \ldots, s_8$, in which the $i$-th queen is in the $i$-th row, and $s_i \in D = \{a, b, c, d, e, f, g, h\}$ specifies its column index. In this way, queen positions in Figure 2 can be encoded as "dbhegacf". The property $F(s_1, \ldots, s_8)$ holds when no two queens can threaten each other. Out of all $8^8$ possible sequences, there are 92 of them that the above property holds.

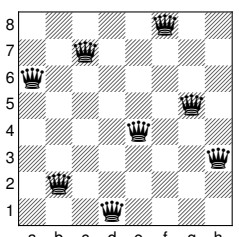

Figure 2: A valid solution of the eight-queen puzzle, sequentially encoded as "dbhegacf".

Researchers have already found that "*cutoff properties*" of the sequential encoding are critical for generating such constrained sequences efficiently (Knuth, 2020). They are a series of properties $F(s_1, \ldots, s_t)$ for $1 \le t < n$, which have the following two characteristics

**Characteristic 4.1** (Inheritability). $F(s_1, \ldots, s_t)$ is true whenever $F(s_1, \ldots, s_{t+1})$ is true;

**Characteristic 4.2** (Incrementality). $F(s_1, \ldots, s_t)$ is fairly easy[2] to test, if $F(s_1, \ldots, s_{t-1})$ holds.

Once a constrained sequence generation task has such properties, it can be solved step-by-step by selecting proper $s_1$ so that $F(s_1)$ holds, proper $s_2$ so that $F(s_1, s_2)$ holds, proper $s_3$ so that $F(s_1, s_2, s_3)$ holds, etc, until all $s_1, \ldots, s_n$ are properly selected. In each step, $s_i$ is selected from a set

$$S_t = \{s \in D \mid F(s_1, \ldots, s_{t-1}, s) \text{ holds}\} \qquad (3)$$

which can be easy to compute given $F(s_1, \ldots, s_{t-1})$ already holds, due to Characteristic 4.2. Such properties admit the concept of "partial (cutoff) candidate solutions" $(s_1, \ldots, s_t), 1 \le t < n$ toward a full solution, which is often much faster than brute-force enumeration of all complete candidates, since it can eliminate many candidates halfway with a single test.

For the aforementioned example of eight-queen puzzle, instead of enumerating all $8^8$ placements to find the attack-free sequences, a much more efficient way is to leverage the cutoff properties of the sequence encoding, in which $F(s_1, \ldots, s_t)$ = "the queens located at $(1, s_1), \ldots, (t, s_t)$ will not attack each other, $1 \le t < 8$". This is fairly easy to test if $F(s_1, \ldots, s_{t-1})$ holds, since we only need to check whether the queen located at $(t, s_t)$ can attack any other queens located at $(1, s_1), \ldots, (t - 1, s_{t-1})$. For example, when $s_1 = d$ and $s_2 = b$, the cutoff property $F(d, b, s_3)$ holds when $s_3 \in S_3 = \{e, g, h\}$. Because cutoff properties reject column and diagonal attacks even on incomplete boards, it examines only 15,720 possible queen placements out of $8^8 = 16,777,216$ to find all valid sequences.

---

**Algorithm 1** Constrained Sequence Generation with Cutoff properties

---

**Input:** Domain $D$, property $F(s_1, \ldots, s_n)$, cutoff properties $F(s_1, \ldots, s_t), 1 \le t < n$.
**Output:** A sequence with $F(s_1, \ldots, s_n)$ holds.
1: $t \leftarrow 1$
2: **while** $t \le n$ **do**
3:     Set $S_t \leftarrow \{s \in D | F(s_1, \ldots, s_{t-1}, s) \text{ holds}\}$
4:     **while** true **do**
5:         **if** $S_t \ne \emptyset$ **then**
6:             Select $s_t$ from $S_t$
7:             $t \leftarrow t + 1$
8:             **break**
9:         **else**       ▷ The backtrack process
10:             $t \leftarrow t - 1$
11:             $S_t \leftarrow S_t \backslash s_t$
12:         **end if**
13:     **end while**
14: **end while**
15: **return** $s_1, \ldots, s_n$

---

---
[2]This is directly quoted from (Knuth, 2020). "Fairly easy" typically means that the test leverages the fact that $F(s_1, \ldots, s_{t-1})$ holds, and only do incremental work to check whether it still holds after adding $s_t$.

| a | ¬a |
|---|---|
| **1** | 0 |
| **0** | 1 |
| **U** | $U$ |

| a ∧ b | | **a** | | |
|---|---|---|---|---|
| | | **1** | **0** | **U** |
| | **1** | 1 | 0 | $U$ |
| **b** | **0** | 0 | 0 | 0 |
| | **U** | $U$ | 0 | $U$ |

| a ∨ b | | **a** | | |
|---|---|---|---|---|
| | | **1** | **0** | **U** |
| | **1** | 1 | 1 | 1 |
| **b** | **0** | 1 | 0 | $U$ |
| | **U** | 1 | $U$ | $U$ |

| a ≃ b | | **a** | | |
|---|---|---|---|---|
| | | **1** | **0** | **U** |
| | **1** | 1 | 0 | 1 |
| **b** | **0** | 0 | 1 | 1 |
| | **U** | 1 | 1 | 1 |

(a) NOT operator      (b) AND operator      (c) OR operator      (d) SIMEQ operator

Table 1: The truth tables for NOT, AND, OR and SIMEQ operators in three-valued logic.

The full algorithm is shown in Algorithm 1. Note that $S_t$ in Equation 3 may be empty, which means a "dead end" at step $t$ that cannot proceed. In this case, we should return back to step $t - 1$, mark $s_{t-1}$ as a dead end, and select another one in $S_{t-1}$ if possible (or repeat the following steps if all of them are dead ends).

## 4.2 A Sequential Representation of Circuits with Cutoff Properties

The main idea of this work is to find a sequential representation of circuits $s_1, s_2, \ldots, s_n$ with aforementioned cutoff properties $F(s_1, \ldots, s_t; f), 1 \leq t < n$ that keep the represented circuit within the equivalence class of $f$. In such a way, we can leverage next-token prediction models to generate tokens step-by-step and validate the logical equivalence in each step, corresponding to line 6 in Algorithm 1. In such a way, the final completed circuit can be guaranteed to be equivalent to $f$.

However, Algorithm 1 include a "backtrack process" in line 10 and 11, which may lead to back-and-forth and is not desired for efficient circuit generation. We note that such a process can be eliminated by assuring $S_t \neq \emptyset$ all the time, leading to the following desired characteristic for the sequential representation:

**Characteristic 4.3** (Backtrack Elimination). *$S_t \neq \emptyset$ is always guaranteed in line 5 of Algorithm 1.*

With such a characteristic, the autoregressive next-token prediction process of circuits can always proceed forward efficiently, analogous to typical natural language generation.

For tasks requiring exploration of feasible regions for circuits, it is important for a representation of circuits to cover the widest possible (ideally all) feasible circuits, minimizing the miss of targets due to the restriction of representation. For example, while we can always generate a strictly equivalent circuit for a Boolean function $f$ via sum-of-product or product-of-sum forms, such forms are too restricted for any feasible region exploration. Therefore, we have the following desired characteristic:

**Characteristic 4.4** (Completeness). *For any $g \in C(f)$, there exists a sequence $s_1, \ldots, s_n$ that uniquely represents $g$.*

Now we propose a sequential representation of circuits with cutoff properties, that has all the aforementioned characteristics.

First, while circuits are usually built in a bottom-up manner from inputs to outputs, we notice that the equivalence constraints are applied on each *output* of the circuit. That is, given the index of output $i$ and a input $\boldsymbol{x}$, a constraint $f_i(\boldsymbol{x}) = g_i(\boldsymbol{x})$ is only possible to be validated when the corresponding circuit output $g_i$ has been built. Therefore, we adopt a special top-down order, specifying a circuit from outputs to inputs, to allow constraint validation throughout the intermediate construction process. It improves sampling efficiency and equivalence preservation of circuit generation.

Then, to allow for indeterminacy in circuit representation, we include the three-valued logic into the circuit evaluation process. That is, besides $\{0, 1\}$ which indicate *false* and *true*, there is another truth value "$U$" which means *unknown*. The truth tables of such logic for NOT, AND and OR operators are shown in Table 1. Additionally, we define a binary operator "SIMEQ" ($\simeq$, is similar or equal to), which is equivalent to the equal operator ($=$) for $\{0, 1\}$, while accommodating $U$ by $U \simeq 0$ and $U \simeq 1$. During the generation process, we relax the equivalence class of $f$ from Equation 2 to

$$C'(f) = \{g \mid g(\boldsymbol{x}) \simeq f(\boldsymbol{x}) \quad \forall \boldsymbol{x} \in \{0, 1\}^N\} \tag{4}$$

so that the occurrence of $U$ in the output will not violate the constraint. For simplicity, here we assume $M = 1$ and leave multi-output cases to be discussed later. The generated circuit $g$ is initialized to be a single constant node $U$, which we call "wildcard node" as it can potentially represent any feasible circuits. So initially, $g(\boldsymbol{x}) \equiv U$ no matter what the input $\boldsymbol{x}$ is. This is within the relaxed feasible region in Equation 4 but does not provide information about the circuit structure.

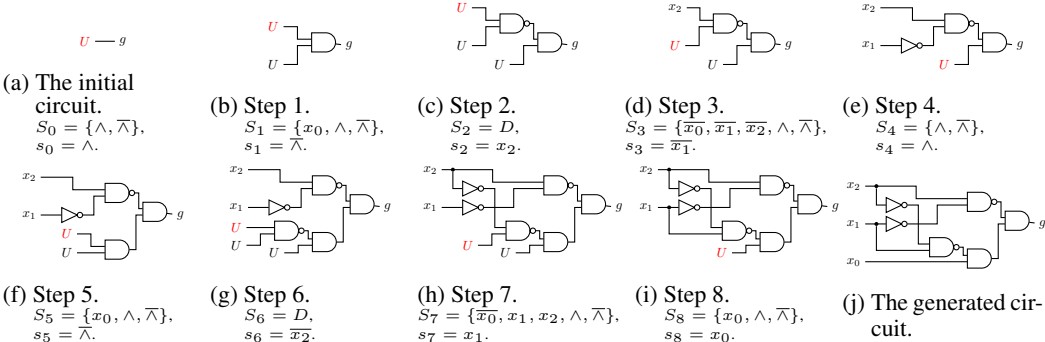

Figure 3: An example showing how a strictly feasible circuit can be built with our proposed sequential representation with cutoff properties. $f(x_2, x_1, x_0)$ is a Boolean function with $f(0, 0, 1) = f(1, 1, 1) = 1$ and $f(x_2, x_1, x_0) = 0$ otherwise. $D = \{x_0, \overline{x_0}, x_1, \overline{x_1}, x_2, \overline{x_2}, \wedge, \overline{\wedge}\}$, $U$ denotes the wildcard node. The next wildcard node to be replaced is marked in red. $S_t = \{s \in D \mid F(s_1, \ldots, s_{t-1}, s; f)$ holds$\}$. The selection of $s_t$ from $S_t$ is based on a masked probability distribution estimated via a trained Circuit Transformer, which will be elaborated in Section 4.4.

Given the initial circuit, the sequential generation process acts as refining the circuit $g$ by recursively replacing a wildcard node to a specific one $s_t \in D$, which can be either a new logic gate or a new primary input. For a new logic gate, all its inputs will be initialized to wildcard nodes that need further refinement. With the proposed top-down approach, the values of $g(\boldsymbol{x})$ for all $\boldsymbol{x} \in \{0, 1\}^M$ in Equation 4 can always be evaluated throughout the construction process, following the truth tables in Table 1. Note that the introduction of three-valued logic also enables short-circuit evaluation with unknown values. For an AND gate $c(\boldsymbol{x}) = a(\boldsymbol{x}) \wedge b(\boldsymbol{x})$, if one of the inputs ($a(\boldsymbol{x})$ or $b(\boldsymbol{x})$) is evaluated to be 0 given specific $\boldsymbol{x}$, then $c(\boldsymbol{x}) = 0$ no matter what the other input is evaluated, even if it is $U$. The same logic applies for the OR gate when one of the inputs is evaluated to be 1.

Then, the cutoff properties $F(s_1, \ldots, s_t; f)$ holds if and only if the circuit partially defined by $s_1, \ldots, s_t$, denoted as $g^{(t)}$, is in the relaxed equivalence class of $f$ in Equation 4. More specifically, for all $\boldsymbol{x} \in \{0, 1\}^M$, the output of the constructed circuit $g^{(t)}$ is similar or equal to $f(\boldsymbol{x})$. That is,

$$F(s_1, \ldots, s_t; f) = \begin{cases} 1, & \text{if } g^{(t)}(\boldsymbol{x}) \simeq f(\boldsymbol{x}), \quad \forall \boldsymbol{x} \in \{0, 1\}^N \\ 0, & \text{otherwise} \end{cases} \quad (5)$$

For the ending criteria, as a wildcard node can potentially be any circuit, only when all the wildcard nodes are recursively replaced by specific ones, can the sequence uniquely represent a circuit, which marks the end of the generation process. For the order of replacement when multiple wildcard nodes exist, we follow a fixed order that prioritizes those with the largest distance from the output and the left child of a gate over the right one. For multi-output cases, we generate the circuit for each output separately, and combine them together via node merging which will be discussed in the next section.

For the selection of logic gates (vocabulary list) in the sequential representation, a standard setting is the combination of AND and NOT gates, which can express all possible truth tables of Boolean functions (termed "functional completeness"), and is also commonly adopted in standard circuit formats such as AIGER (Biere, 2007). Therefore, we adopted this setting. Additional types of gates can be further included to adapt to different circuit settings, which is left as a future work. An alteration is that we merged the NOT gate with the AND gate and primary inputs. Instead of assigning the NOT gate an individual token, each primary inputs and the AND gate has two versions: the original ones ($x_1, \ldots, x_N, \wedge$) and the inverse ones ($\overline{x_1}, \ldots, \overline{x_N}, \overline{\wedge}$), so the vocabulary list contains $2N + 2$ tokens in total[3]. That is,

$$D = \{x_1, \overline{x_1}, \ldots, x_N, \overline{x_N}, \wedge, \overline{\wedge}\}. \quad (6)$$

This allows us to significantly shorten the sequence with a moderate increase of vocabulary size.

An example of our proposed representation and cutoff properties are shown in Figure 3. We leave the proof of Characteristic 4.1, 4.2, 4.3 and 4.4 in Section A.2. Given Equation 3, Equation 5 and Characteristic 4.2, we can compute $S_t$ in a time complexity of $O(N \cdot 2^N \cdot d)$. $d$ is the depth of the wildcard node to replace at step $t$. The detail is leaved in Section A.3.

---

[3]This does not include special tokens such as [EOS] and [PAD] in Transformer models.

### 4.3 FROM TREES TO DIRECTED ACYCLIC GRAPHS

In the previous section, we proposed a sequential representation of circuits based on recursive replacement of wildcard nodes in a top-down manner. Such an approach implicitly assumes that a logic gate would always have a single outgoing edge, restricting the generated circuits to be highly hierarchical with tree structures. However, multi-fanout gates do commonly exist in real-world logic circuits, which shape circuits as directed acyclic graphs (DAGs).

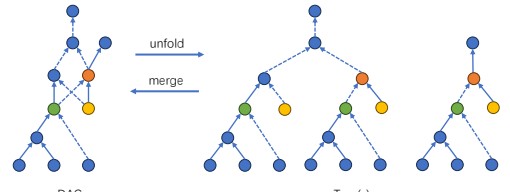

Figure 4: Transition between a DAG and one or more trees.

In this section, we show how we extend our method to generate DAG circuits. We notice that a DAG can be "unfolded" to one or more trees once we duplicate every node with outdegree larger than one, so that every node has at most one outgoing edge. For example, the orange node in the left DAG of Figure 4 is duplicated into two individual nodes, where its two outgoing edges are assigned respectively. Reversely, one or more trees can also be transformed to a DAG by merging nodes with structural equivalence. For circuits, such a bidirectional transition will not change the Boolean function it represents. Therefore, we generate a DAG circuit by firstly generating its unfolded tree representation, and then merging equivalent nodes in the generated tree representation. In logic circuit design, different nodes can be not only structurally equivalent but also functionally equivalent (Mishchenko et al., 2005), which means that their outputs represent the same Boolean functionality. Functionally equivalent nodes can thus be merged as a single node even if they have different underlying structures. Our approach mainly leverages the functional equivalence.

### 4.4 NEURAL ENCODING OF CIRCUITS AND CIRCUIT TRANSFORMER

While we can deserialize our proposed sequential representation to a DAG circuit via node merging, we can also serialize a given DAG circuit to our proposed sequential representation in a similar manner, via a depth-first traversal with node duplication. Given a DAG circuit, we start a traversal from each of its primary outputs, and visit each connected gate in a depth-first and recursive manner. Backtracking occurs when a primary input is reached. Importantly, such a traversal is memoryless, i.e., visited nodes will not be labelled during the traversal, thus a node will appear multiple times in the trajectory if its fan-out is larger than one, corresponding to the node duplication in the previous section. When the process is finished, the traversal trajectory $s_1, \ldots, s_n$ is the sequential representation of the unfolded tree version of the original DAG circuit. Note that such an unfolding process may lead to long sequences, especially for nodes with large number of outgoing edges. A more compact representation is left as future work.

For Transformer models to process the sequential representation, it is important to provide an efficient positional encoding for each node to indicate its position in the circuit. In this work, we utilize the path from the primary output to a given node to indicate the node's position. To achieve this, we follow (Shiv & Quirk, 2019) that encodes the path as a stack of one-hot encodings ("10" for the first input and "01" for the second input). More details are left in the appendix.

With all the circuit encoding and generation techniques introduced above, we propose the Circuit Transformer, an end-to-end Transformer model that generates a functionally equivalent circuit of a Boolean function. The Transformer adopts the classical encoder-decoder Transformer architecture (Vaswani et al., 2017). The Boolean function to be implemented is specified as a circuit, and encoded by the Transformer encoder with the aforementioned encoding approach. The Transformer decoder adopted the mechanism in Section 4.2 by inserting a masking layer before the final softmax layer, which reflects $S_t$. When predicting the next token $s_{t+1}$, the masking layer blocks candidates that invalidate equivalence, based on the current "cutoff" circuit partially defined by $s_1, \ldots, s_t$. The Transformer is trained in a standard supervised way, which minimize the cross entropy between the predicted distribution and the ground truth. Moreover, node merging proposed in Section 4.3 is applied to the decoding process on-the-fly to transform the unfolded tree representation to a DAG circuit, whose details can be found in Algorithm 3 in the appendix. The training and inference stage is shown in Figure 5. We denote a trained Circuit Transformer as

$$P_{\text{CT}}(s \mid s_1, \ldots, s_t; f)$$

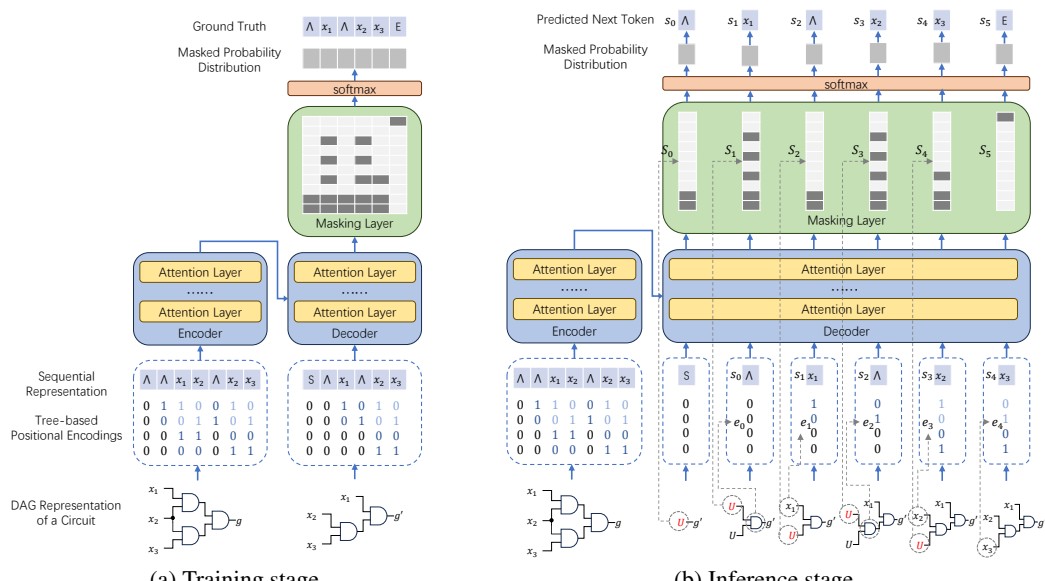

(a) Training stage                    (b) Inference stage

Figure 5: The training and the inference stage of the Circuit Transformer. In the inference stage, the positional encoding $e_t$ at step $t$ is computed on-the-fly based on the position of the current token $s_t$ in the partially defined circuit. The mask $S_{t+1}$ for predicting the next token $s_{t+1}$ is computed based on Equation 3 given the previous tokens $s_1, \ldots, s_t$ and the cutoff property $F(s_1, \ldots, s_t; f)$. "S" and "E" denote the [START] and [END] tokens in Transformer model.

in which $\sum_{s \in S_{t+1}} P_{\mathrm{CT}}(s \mid s_1, \ldots, s_t; f) = 1$ and $P_{\mathrm{CT}}(s \mid s_1, \ldots, s_t; f) = 0, \forall s \in D \backslash S_{t+1}$. The input circuit specifying $f$ is omitted for simplicity.

## 4.5 EQUIVALENCE-PRESERVING CIRCUIT OPTIMIZATION AS A MARKOV DECISION PROCESS

An important application of equivalent circuit generation is to optimize circuits with respect to certain objectives. For example, one important problem is to find a compact implementation of a Boolean function $f$ with minimal number of logic gates (the circuit size minimization problem). That is

$$\min |g| \quad \text{s.t.} \quad g \in C(f), \tag{7}$$

where $|g|$ is the number of logic gates (NOT gates are not counted to align with the mainstream research setting) in $g$. Under our proposed sequential representation, it can be achieved by attaching an immediate reward function $R(s_1, \ldots, s_t, s)$ to the generation of token $s$ at step $t$, so that the sum of the reward function throughout the generation reflects the objective. In this way, the generation process can be regarded as a deterministic Markov decision process, in which the state at step $t$ is the generated tokens $(s_1, \ldots, s_t)$, the set of actions available from the state is $S_t$ in Equation 3, the immediate reward is $R(s_1, \ldots, s_t, s)$, and the next state is $(s_1, \ldots, s_t, s)$ with probability 1. The process terminates once $(s_1, \ldots, s_t)$ represents a unique circuit with all wildcard nodes replaced.

Such a formulation has two key advantages. First, the feasibility of the generated circuit is guaranteed once the process terminates. No effort is required to penalize infeasible cases via crafting the reward function. Second, the size of the available action set is at most $2N + 2$, in which $N$, the number of inputs, is usually moderately small in practice. Other circuit representations typically assign a unique ID to each logic gate to describe their adjacency, requiring the size of available actions to be proportional to the number of gates, which is usually significantly larger than $N$.

For the circuit size minimization problem in Equation 7, the immediate reward function can be defined as

$$R(s_1, \ldots, s_t, s) = \Delta + \begin{cases} -1, & s = \wedge \text{ or } s = \overline{\wedge} \\ 0, & \text{otherwise} \end{cases} \tag{8}$$

in which $\Delta$ reflects the refinement due to equivalent node merging. Given a depth-first replacement order of wildcard nodes in Section 4.2, the node merging process in Section 4.3 can be done simultaneously with the generation process, whose details are left in the appendix.

Given the reward function, we can adopt Monte-Carlo tree search (MCTS) to maximize the cumulative reward $\sum_t R(s_1, \ldots, s_t)$ while strictly preserving the equivalence. Each search node will be "simulated" once to obtain a value $v$, which is the estimated cumulative reward when choosing the path from the root node to this node. Each node also stores four values: the average value $Q(s)$, the number of visits $N(s)$, the immediate reward $R(s)$ and the prior probability $P(s)$. When selecting a child $a$ for the node $s$, MCTS make a balance between exploitation and exploration by assigning each child a PUCT score (Silver et al., 2017) as follows

$$\text{PUCT}(a) = Q(a) + cP(a)\frac{\sqrt{N(s)}}{1 + N(a)} \tag{9}$$

in which $c$ is a hyperparameter (set to 1 in our case). $P(a)$ is a prior probability of selecting $a$, allowing nodes with larger $P(a)$ obtain higher score. We adopt Circuit Transformer to estimate $P(a)$ as well as conducting simulation to get the value $v$. The detailed procedure is shown in Algorithm 4 in the appendix.

## 5 EXPERIMENTS

In this section, we supervisedly train a Circuit Transformer to solve the circuit size minimization problem in Equation 7, generating equivalent yet more compact forms of input circuits, and conduct extensive experiments on both synthetic and real datasets to evaluate its feasibility and optimality.

We conduct the experiments on 8-input, 2-output circuits, which can specify $(2^{2^8})^2 = 1.34 \cdot 10^{154}$ different Boolean functions. Optimizing a circuit while exactly matching one of the functions is challenging. Such a size is out of the capacity of current exact solvers and significantly larger than the typical sub-circuit size (4-6 inputs and one output) for traditional divide-and-conquer methods. Effective end-to-end optimization on such a circuit size leads to enhancement of global optimality for large circuit optimization (Li & Dubrova, 2011; Zhu et al., 2023).

The detailed parameter setting of the Circuit Transformer model are as follows. The embedding width and the size of feedforward layer are set to 512 and 2048 following (Vaswani et al., 2017), while the number of attention layers is set to 12, leading to 88.2 million total parameters, a moderate size that allows efficient training and evaluation on a single, customer-grade GPU. The vocabulary size is 20 (8 inputs and the AND gate, with their inverse, plus [EOS] and [PAD]). Batch size is set to be 128. The maximum length of the input and output sequence is set to be 200. To evaluate the effectiveness of tree positional encoding (TPE) in Section 4.4, we trained Circuit Transformers with and without TPE. The maximal depth of tree positional embeddings is set to be 32. The implementation is based on (Yu et al., 2020). During inference, the prediction of tokens from distributions is deterministic (the token with the largest probability is selected) for reproducibility.

We also trained two baseline Transformer models with exactly the same experimental settings, except employing different sequential representation of circuits as follows:

- Boolean Chain (Knuth, 2015): a representation that is extensively applied in SAT-based optimization techniques. A chain is initialized by all the primary inputs of the circuit, and each gate is represented by two prior indices in the chain, indicating the source of its two inputs.
- AIGER (Biere, 2007): a popular representation of logic circuits with AND and NOT gates. We follow the tokenization setting in (Schmitt et al., 2021), representing an AND gate by three tokens followed by a special new line token.

To train and evaluate the Circuit Transformer model, we build a large dataset containing 15 million randomly generated 8-input, 2-output circuits. The supervised signals (i.e., the equivalent circuits that are optimized in size) are generated by the Resyn2 command in ABC (Brayton & Mishchenko, 2010), a representative optimization flow for circuit size minimization. The detail of random circuit generation is left in the appendix. 89% of the data is for training, 1% is for validation and 10% is reserved for testing. All the Transformer models are trained on the training set sufficiently for 5 epochs on a single NVIDIA GeForce RTX 4090 graphic card for 75 hours.

We employ both a synthetic dataset and real EDA benchmarks to evaluate the performance:

- Random circuits: 10,240 circuits are randomly sampled from the test set of the aforementioned randomly generated dataset. The average size is $25.83 \pm 5.38$.

| Methods | Random circuits | | IWLS FFWs | |
|---|---|---|---|---|
| | Unsuccessful cases | Avg. size | Unsuccessful cases | Avg. size |
| Boolean Chain | 5.07% (5.07%) | 15.25 | 11.36% (11.26%) | 17.24 |
| Boolean Chain (beam size = 16) | 2.16% (2.16%) | 14.89 | 6.34% (6.29%) | 17.15 |
| Boolean Chain (beam size = 128) | 1.91% (1.91%) | 14.87 | 5.97% (5.94%) | 17.15 |
| AIGER | 4.32% (4.32%) | 15.14 | 8.35% (7.77%) | 17.19 |
| AIGER (beam size = 16) | 1.85% (1.85%) | 14.87 | 4.62% (4.37%) | 17.12 |
| AIGER (beam size = 128) | 1.71% (1.71%) | 14.86 | 4.24% (3.99%) | 17.12 |
| Circuit Transformer w/o TPE | 2.14% (0%) | 15.02 | 6.63% (0%) | 17.33 |
| Circuit Transformer | 1.14% (0%) | 14.79 | 4.76% (0%) | 17.17 |
| Circuit Transformer ($K = 10$) | 0.20% (0%) | 14.02 | 2.83% (0%) | 16.92 |
| Circuit Transformer ($K = 100$) | **0.17%** (0%) | **13.73** | **2.63%** (0%) | **16.73** |
| Resyn2 (ground truth for training) | / | 14.56 | / | 16.82 |

Table 2: Results on 10,240 randomly generated circuits, and 10,240 fanout-free windows randomly sampled from the IWLS 2023 benchmark. For unsuccessful cases, the percentage in the bracket corresponds to failures due to equivalence constraint violation. All results of Circuit Transformers show zero violation of equivalence constraints. $K$ denotes the total number of playouts in Monte-Carlo tree search. All the models are supervisedly trained on the Resyn2 optimized circuits.

- IWLS FFWs: we transform the IWLS 2023 benchmark (Mishchenko, 2023) into circuits represented by AND and NOT gates by the script suggested in (Mishchenko & Chatterjee, 2022), and extract 1.5 million 8-input, 2-output fanout-free windows (FFWs), a kind of sub-structure of large circuits (Zhu et al., 2023). Then we randomly sample 10,240 circuits from the extracted FFWs. The average size is $18.01 \pm 6.47$.

To enhance the performance of the Transformer models in comparison, two heuristics search techniques are applied. Monte-Carlo tree search in Algorithm 4 is applied for Circuit Transformer to evaluate our proposed MDP formulation in Section 4.5. For Transformer models trained on Boolean chain and AIGER, applying MCTS is significantly more inefficient due to the large set of available actions discussed in Section 4.5, so we adopt Beam search as an alternative, which finds the most probable sequence by maintaining a fixed size (beam size) of candidates.

The results are presented in Table 2, with a more detailed version in Section A.9. On both synthetic and real datasets, the Circuit Transformer surpasses two other Transformer models in terms of feasibility (measured by the percentage of unsuccessful cases) and optimality (measured by the average circuit size). The two baseline models generate unsuccessful circuits for various reasons, including equivalence constraint violations, cycles in circuits, or incomplete specifications, with the most common issue being that the generated circuit is complete and valid but not strictly equivalent to the original. In contrast, the Circuit Transformer's exact precision is empirically shown by zero violation of complex equivalence constraints. The only reason for unsuccessful cases is that wildcard nodes are not fully replaced within the given maximum sequence length of 200. Case studies can be found in the appendix. Regarding heuristic search enhancement, while beam search significantly improved feasibility, consistent with findings in (Schmitt et al., 2021), the issue of non-equivalence remained prevalent. Conversely, Monte-Carlo tree search in the Circuit Transformer not only substantially reduced unsuccessful cases but also significantly improved the average circuit size, sometimes producing circuits more compact than the ground truth with a moderate number of playouts.

# 6 CONCLUSION

In this work, we make an important advancement towards achieving precise generative AI for logic tasks, demonstrating that complex hard-constraint satisfaction is attainable for next-token prediction models when a proper formulation of the constrained problem is established. Inspired by the cutoff properties, we introduce a novel approach to the fundamental problem of equivalent circuit generation, enabling next-token prediction models to generate new logic circuits while strictly adhering to complex equivalence constraints. Future works includes extending such methodology to other fundamental constrained problems, integrating current models into industrial large circuit optimizers, and exploiting more compact representation of circuits such as BDDs.

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

# A APPENDIX

## A.1 DETAIL OF BACKTRACKING FRAMEWORK

The basic backtracking algorithm is as follows, extracted from (Knuth, 2020):

Given domain $D$ and properties $F(s_1, \ldots, s_t)$, this algorithm visites all sequence $s_1, s_2, \ldots, s_n$ that satisfy $F(s_1, \ldots, s_n)$:

Step 1 [Initialize] Set $t \leftarrow 1$, and initialize the data structures needed later.
Step 2 [Enter level $t$] (Now $F(s_1, \ldots, s_{t-1})$ holds.) If $t > n$, visit $s_1, \ldots, s_n$ and go to Step 5, Otherwise set $s_t \leftarrow \min D$, the smallest element of $D$.
Step 3 [Try $s_t$] If $F(s_1, \ldots, s_t)$ holds, update the data structures to facilitate testing $F(s_1, \ldots, s_t, s_{t+1})$, set $t \leftarrow t + 1$, and go to Step 2.
Step 4 [Try again] If $s_t \neq \max D$, set $s_t$ to the next larger element of $D$ and return to Step 3.
Step 5 [Backtrack] Set $t \leftarrow t - 1$. If $t > 0$, downdate the data structures by undoing the changes recently made in Step 3, and return to Step 4. (Otherwise stop.)

We refer to Section 7.2.2 of (Knuth, 2020) for more details about backtracking.

## A.2 PROOF OF CHARACTERISTICS

In this section, we demonstrate that our proposed sequential representation has Characteristic 4.1, 4.2, 4.3 and 4.4.

Characteristic 4.1: When $F(s_1, \ldots, s_{t+1}; f)$ is true, the transition from $s_1, \ldots, s_{t+1}$ to $s_1, \ldots, s_t$ corresponds to reversely replacing a specific node $s_{t+1}$ with a wildcard node. such a replacement will never break the feasibility, because (1) a wildcard node only represents feasible circuits; (2) the wildcard node at least have a feasible choice to be set as $s_{t+1}$ as $s_1, \ldots, s_{t+1}$ is feasible.

Characteristic 4.2: During the generation from step 1 to step $t - 1$, a cache mechanism can be employed to cache the truth table of the constructed nodes. Therefore, when $F(s_1, \ldots, s_{t-1}; f)$ holds and $F(s_1, \ldots, s_{t-1}, s_t; f)$ needs to be evaluate, we simply traverse the generated circuit in a bottom-up manner, from the current wildcard node to be replaced to the root, to evaluate $g^{(t)(x)}$ for all $x \in \{0, 1\}^N$ with time complexity of $O(N \cdot 2^N \cdot d)$ in which $d$ is the depth of the node. More details about the cache mechanism can be found in Section A.3.

Characteristic 4.3: Note that the AND gate $\wedge$ and the NAND gate $\overline{\wedge}$ is always in $S_t$ in our sequential representation, as replacing a wildcard node to an AND or NAND gate with two wildcard nodes will never break the feasibility.

Characteristic 4.4: For all $g \in C(f)$, the sequence $s_1, \ldots, s_n$ that represents $g$ is demonstrated in Section 4.4.

## A.3 THE COMPUTATION OF $S_t$

In Section 4.2, we need to compute

$$S_t = \{s \in D | F(s_1, \ldots, s_{t-1}, s; f) \text{ holds}\}$$

in which

$$F(s_1, \ldots, s_t; f) = \begin{cases} 1, & \text{if } g^{(t)}(x) \simeq f(x), \quad \forall x \in \{0, 1\}^N \\ 0, & \text{otherwise} \end{cases}$$

Literally, such a computation is done by initializing $S_t$ as $\emptyset$, and iterating through all possible symbols $s$ in $D$ to see whether it should be added to $S_t$. For each $s$, we do the following:

1. Append $s$ to $s_1, \ldots, s_{t-1}$ (i.e., replace a wildcard node of current partial circuit by $s$, to form a new partial circuit $g^{(t)}$).
2. Iterate through all possible inputs $x \in \{0, 1\}^N$ to see if any equivalence conflict (i.e. $g^{(t)}(x) = 0$ and $f(x) = 1$, or $g^{(t)}(x) = 1$ and $f(x) = 0$) occurs in the new partial circuit $g^{(t)}$.
3. If we can find any $x$ that leads to an equivalence conflict, $s$ should not be added to $S_t$. If there is no equivalence conflict for all $x \in \{0, 1\}^N$, $s$ should be added to $S_t$.

However, the above process is not very efficient in practice for two reasons. First, many nodes in the current partial circuit are already "fully constructed" or "fixed", which means that their truth table only contains 0 and 1, which will no longer change during the generation process. However, in step 2, such nodes with their child nodes still participate in the calculation of $g^{(t)}(\boldsymbol{x})$ for every $\boldsymbol{x} \in \{0,1\}^N$, leading to unnecessary repetitiveness. Second, the calculation of $g^{(t)}(\boldsymbol{x})$ for a single $\boldsymbol{x} \in \{0,1\}^N$ is performed on a DAG, which cannot benefit from the acceleration of matrix computation.

In this section, we introduce how to compute $S_t$ in $d$ steps of matrix-vector computation (with a time complexity of $O(N \cdot 2^N \cdot d)$), in which $d$ is the depth of the wildcard node to replace at step $t$. In such a way, the computation will not be a bottleneck when $N$ is reasonably small (in our case $N = 8$). First, during the construction process, we name a node $s$ in a circuit as "fixed" if the value of $s$ is either 0 or 1 (not $U$) for all possible inputs. Formally,

$$s \text{ is fixed iff } s(\boldsymbol{x}) \neq U, \forall \boldsymbol{x} \in \{0,1\}^N$$

in which $s(\boldsymbol{x})$ is the value of node $s$ given input $\boldsymbol{x}$. Therefore, an input node is always fixed. An intermediate node must be fixed if its left and right child are both fixed. For example, in the partial sequence $[\wedge \wedge x_1 x_2]$, the second $\wedge$ is fixed because its left and right children are $x_1$ and $x_2$, which are both fixed. The first $\wedge$ is not fixed because its right child is $U$, its value is $U$ when $x_1$ and $x_2$ are both 1 ($1 \wedge U = U$).

Recall that during the construction process in Section 4.2, for the order of replacement when multiple wildcard nodes exist, we follow a fixed order that prioritizes those with the largest distance from the output, and prioritizes the left child of a gate over the right one if they have the same distance. Therefore, the construction process will only replace a node's right wildcard child if there is no wildcard nodes in its left branch. In other words, in the construction process, when the wildcard node to replace is $u$, then for any node $s$ on the path from $u$ to the root,

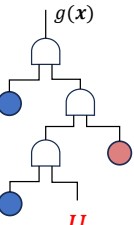

$g(\boldsymbol{x})$

$U$

- If $u$ is in the right branch of $s$, the left child of $s$ must be fixed.
- If $u$ is in the left branch of $s$, the right child of $s$ must be a wildcard node.

An illustration is shown in Figure 6. Therefore, we can cache the truth table of the fixed nodes (the blue nodes in Figure 6) by maintaining a cache dictionary $d$ during the construction process. When

Figure 6: When the wildcard to replace is the node $U$ in red, the blue nodes on the left must be fixed, and the red nodes on the right must be $U$.

computing $g^{(t)}(\boldsymbol{x})$ for all $s \in D$ and $\boldsymbol{x} \in \{0,1\}^N$, we first initialize the wildcard node as an $2N \times 2^N$ Boolean matrix $B$, each row storing the truth table of $x_1, \overline{x_1}, \ldots, x_N, \overline{x_N}$. Then, we backtrack from the wildcard node $U$ to the root in Figure 6, and do the following computation in each step

- If the current node $u$ is the right child of its parent, query the cache $d$ for the truth table $d[l]$ of the left child $l$ (a blue node), and update $B[i] \leftarrow B[i] \wedge d[l]$ for each row of $B$ (can be efficiently done by broadcasting mechanism[4].
- If the current node $u$ is the left child of its parent, the right child must be $U$ (a red node), so we simply update $B \leftarrow B \wedge U$ in an element-wise way.
- If the current node $u$ is $\overline{\wedge}$, update $B$ by its inverse. Then update $u$ by its parent to start the next step.

When we finished the backtrack process and reached the root, $B$ will store the truth table of $g^{(t)}(\boldsymbol{x})$ for every $s \in \{x_1, \overline{x_1}, \ldots, x_N, \overline{x_N}\}$. Then we simply compare each row of $B$ with the truth table of $f$ to see if there is any equivalence conflict, and add tokens without conflict in $S_t$. $\wedge$ and $\overline{\wedge}$ will never cause a conflict, so they will always stay in $S_t$. The detailed process is shown in Algorithm 2.

### A.4 DEMONSTRATION OF THE TREE POSITIONAL ENCODING

For example, in Figure 5, the position of the second node in the sequence, i.e., the uppermost AND gate connecting $x_1$ and $x_2$, can be represented as $e_2 = [10]$ (this gate's output is the first input of the

---

[4]https://numpy.org/doc/stable/user/basics.broadcasting.html

---

**Algorithm 2** The Computation of $S_t$

---

**Input:** The truth table $t$ of the Boolean function $f$ to be realized, the wildcard node $u$ to be replaced at time step $t$, a cache dictionary $d$ of fixed nodes storing their truth tables.
**Output:** $S_t$
  $S_t \leftarrow \{\wedge, \overline{\wedge}\}$
  $b \leftarrow$ the truth tables of input nodes $x_1, \overline{x_1}, \ldots, x_N, \overline{x_N}$
  **while** $u$ is not root **do**
    **if** $u$ is the left node of its parent **then**
      Set $b[s] \leftarrow b[s] \wedge U \quad \forall s \in \{x_1, \overline{x_1}, \ldots, x_N, \overline{x_N}\}$
    **else**
      Set $b[s] \leftarrow b[s] \wedge d[l] \quad \forall s \in \{x_1, \overline{x_1}, \ldots, x_N, \overline{x_N}\}$, in which $l$ is the left node of $u$'s parent
    **end if**
    **if** $u = \overline{\wedge}$ **then**
      Set $b[s] \leftarrow \neg b[s] \quad \forall s \in \{x_1, \overline{x_1}, \ldots, x_N, \overline{x_N}\}$
    **end if**
  **end while**
  **for** $s \in \{x_1, \overline{x_1}, \ldots, x_N, \overline{x_N}\}$ **do**
    **if** there is no zero in $b[s] \simeq t$ **then**
      Add $s$ to $S_t$
    **end if**
  **end for**
  **rturn** $S_t$

---

rightmost AND gate, so "10" is pushed in the encoding stack of the rightmost AND gate, which is empty), and the position of the fourth node $x_2$ in the sequence can be represented as $e_4 = [01; e_2] = [0110]$ (push "01" in $e_2$ as $x_2$ is the second node's second input) and $e_6 = [10; e_5] = [1001]$ when $x_2$ is secondly visited as the fifth node's first input.

For circuits with multiple primary outputs ($M > 1$), we initialize the encoding stack of each primary output with a unique one-hot encoding, as if there is a virtual root node of $M$ children, and each primary output corresponds to one of the children.

### A.5 IMMEDIATE EQUIVALENT NODE MERGING AND FUNCTIONAL EQUIVALENCE CHECKING

With a depth-first replacement order, we can follow Algorithm 3 to merge equivalent nodes during the generation process. For functional equivalence checking of two nodes $p$ and $q$, we check whether $p(\boldsymbol{x}) = q(\boldsymbol{x}), \forall \boldsymbol{x} \in \{0, 1\}^N$ by iterating all $\boldsymbol{x}$. If there is an $\boldsymbol{x}$ such that $p(\boldsymbol{x}) \neq q(\boldsymbol{x})$, then $p$ and $q$ are not functionally equivalent.

### A.6 CIRCUIT TRANSFORMER WITH MONTE-CARLO TREE SEARCH

The detailed procedure is shown in Figure 4.

### A.7 DATASET GENERATION

The process to generate a random circuit is shown in Algorithm 5. We restrict that the length of the encoded sequence for each circuit should fit all the three sequential representations with a maximal length of 200, and all the 8 inputs should appear in the circuit. Each circuit has a unique structure, which is realized by a canonicalization technique (Chai & Kuehlmann, 2006).

### A.8 EXPERIMENTS COMPUTE RESOURCES

All the experiments are conducted on a workstation with the following specification:

- CPU: AMD Ryzen™ 9 7950X Desktop Processor (16 cores, 32 threads)
- Memory: 192GB (48GB × 4) DDR5 5200MHz
- GPU: NVIDIA GeForce RTX 4090 × 2

Each Transformer model in the experiments is trained on a single GPU with 75 hours.

---

**Algorithm 3** Circuit generation with immediate equivalent node merging

---

**Input:** The Boolean function $f$ that the generated circuit should be equivalent to. Next token prediction model $P(s_t|s_1, \ldots, s_{t-1})$.
**Output:** A feasible circuit $g$ satisfying $g \in C(f)$.
1: Initialize $path$ as an empty stack of gates, $POs$ as an empty list.
2: Initialize $G = \emptyset$ as a set of non-isomorphic gates
3: **for** $t = 1, 2, 3, \ldots$ **do**
4:  Compute a probability distribution of $s_t \in D$ by the next token prediction model

$$p_t \leftarrow P(s_t|s_1, \ldots, s_{t-1})$$

5:  Set $S_t \leftarrow \{s \in D|F(s_1, \ldots, s_{t-1}, s; f) \text{ holds}\}$    $\triangleright$ $S_t \neq \emptyset$ is guaranteed by Characteristic 4.3
6:  $s_t \leftarrow \arg\max_{s \in S_t} p_t(s)$
7:  Initialize $s_t.input1 \leftarrow U, s_t.input2 \leftarrow U$ if $s_t$ is a gate.
8:  **if** $path$ is empty **then**          $\triangleright$ the output of $s_t$ is the primary output of the circuit
9:    Append $s_t$ to $POs$ and push $s_t$ to $path$
10:  **else**              $\triangleright$ $s_t$ should be the input of the last gate in $path$
11:    $s \leftarrow path.peek()$           $\triangleright$ get the last gate added to $path$
12:    **if** $s.input1 = U$ **then** $s.input1 \leftarrow s_t$ **else** $s.input2 \leftarrow s_t$  $\triangleright$ replace a wildcard node in $s_t$ to $s$
13:    **if** $s_t$ is a gate **then**
14:      $path.push(s_t)$
15:    **else**    $\triangleright$ $s_t$ is an input node in $x_1, \overline{x_1}, \ldots, x_N, \overline{x_N}$. Pop fully constructed gates from $path$
16:      **while** $s.input1 \neq U$ **and** $s.input2 \neq U$ **do**
17:        **if** $s \in G$ **then**    $\triangleright$ Compute the truth table of $s$ to check functional equivalence
18:          Update $path$ and $POs$ to replace $s$ with the functional equivalent one in $G$
19:        **else**
20:          Add $s$ to $G$
21:        **end if**
22:        $s \leftarrow path.pop()$
23:      **end while**
24:    **end if**
25:  **end if**
26: **end for**
27: Return the circuit with POs as $POs$

---

## A.9   DETAILED EXPERIMENTAL RESULT

More detailed results are shown in Table 3 and Table 4. For all unsuccessful cases, the size of optimized circuit is regarded as the size of the original circuit (i.e., the model does nothing). For all the results regarding Circuit Transformer, the time cost of masking layer contributes to 6% of the total time cost. Note that the decrease of time cost by removing the masking layer is more significant than 6%, due to the fact that the masking layer "forces" the model to generate longer sequences to satisfy the constraint, which costs more time. Also note that the size of the Transformer model significantly impacts the time cost, and the reported time cost only reflects the current size setting of 88 million parameters. The time cost with heuristics search is generally proportional to the beam size / search rounds.

## A.10   CASE STUDY

To show how the unsuccessful cases looks like for both the baselines and Circuit Transformer, Table 5 shows three categories of unsuccessful cases for the baseline Transformer model trained on AIGER format, and provide an example from the IWLS FFWs dataset for each category. Table 6 shows the only circumstance that Circuit Transformer is unsuccessful, which is the exceeding of the pre-set maximal sequence length of 200.

**Algorithm 4** Circuit Generation with Circuit Transformer and Monte-Carlo Tree Search

---

**Input:** The Boolean function $f$ to be realized. Circuit Transformer $P_{\text{CT}}(s_t|s_1, \ldots, s_{t-1}; f)$. Immediate reward function $R(s_1, \ldots, s_t)$. Number of playouts $K$.
**Output:** An improved sequential representation of circuit
  Create a root search node $r$, initialize $v_{\max}$ as a small number, $BestSeq \leftarrow []$.
  **for** $i = 1, 2, \ldots, K$ **do**
    Starting from $r$, iteratively selects a child node with the largest PUCT score, until reaching a leaf node $l$.
    Set $v \leftarrow R(r) + \cdots + R(l)$
    **if** $l$ has been simulated before **then**
      Expand $l$ by adding all valid tokens $\{s \in D \mid F(r, \ldots, l, s; f) \text{ holds}\}$ as its children
      Compute prior probability $P(a)$ for each child $a$ of $l$ via $P(a) \leftarrow P_{\text{CT}}(a \mid r, \ldots, l; f)$
      Initialize $Q(a) \leftarrow 0, N(a) \leftarrow 0$ for each child $a$ of $l$
      Select the child $c$ with the largest PUCT score and set $R(c) \leftarrow R(r, \ldots, l, c), v \leftarrow v + R(c)$
    **else**
      Set $c \leftarrow l$
    **end if**
    **while** $c \neq \text{EOS}$ **do**
      Set $c \leftarrow \arg\max_s P_{\text{CT}}(s \mid r, \ldots, c; f)$, and then set $v \leftarrow v + R(r, \ldots, c)$
    **end while**
    **if** $v > v_{\max}$ **then** $v_{\max} \leftarrow v, BestSeq \leftarrow [r, \ldots, c]$
    **for** $s = c, \ldots, r$ **do**
      Set $Q(s) \leftarrow (Q(s) \cdot N(s) + v)/(N(s) + 1), N(s) \leftarrow N(s) + 1$
    **end for**
  **end for**
  **Return** $BestSeq$

---

Table 3: Detailed results for random circuits.

| Methods | Random Circuits | | | | | |
| --- | --- | --- | --- | --- | --- | --- |
| | Unsuccessful cases | Violation cases | Average circuit size | SD of circuit size | % of circuits strictly smaller than Resyn2 | Average time per circuit (s) |
| Boolean Chain | 5.07% | 5.07% | 15.25 | 4.52 | 15.49% | 0.010 |
| Boolean Chain (beam size=16) | 2.16% | 2.16% | 14.89 | 3.86 | 16.42% | 0.048 |
| Boolean Chain (beam size=128) | 1.91% | 1.91% | 14.87 | 3.82 | 16.57% | 0.369 |
| AIGER | 4.32% | 4.32% | 15.14 | 6.34 | 16.22% | 0.018 |
| AIGER | 1.85% | 1.85% | 14.87 | 3.78 | 16.06% | 0.097 |
| AIGER | 1.71% | 1.71% | 14.86 | 3.73 | 16.25% | 0.734 |
| Circuit Transformer w/o Masking | 2.59% | 2.59% | 14.95 | 3.83 | 15.49% | 0.010 |
| Circuit Transformer w/o TPE | 2.14% | 0.00% | 15.02 | 3.83 | 16.44% | 0.018 |
| Circuit Transformer | 1.14% | 0.00% | 14.79 | 3.48 | 16.84% | 0.018 |
| Circuit Transformer ($K$=10) | 0.20% | 0.00% | 14.02 | 2.79 | 36.15% | 0.210 |
| Circuit Transformer ($K$=100) | **0.17%** | 0.00% | **13.73** | 2.62 | **46.77%** | 2.090 |
| Resyn2 (ground truth for training) | 0.00% | 0.00% | 14.56 | 2.98 | 0.00% | 0.009 |

Table 4: Detailed results for IWLS fanout-free windows.

| Methods | IWLS FFWs | | | | | |
| --- | --- | --- | --- | --- | --- | --- |
| | Unsuccessful cases | Violation cases | Average circuit size | SD of circuit size | % of circuits strictly smaller than Resyn2 | Average time per circuit (s) |
| Boolean Chain | 11.36% | 11.27% | 17.24 | 6.40 | 7.71% | 0.015 |
| Boolean Chain (beam size=16) | 6.34% | 6.29% | 17.15 | 6.29 | 8.42% | 0.079 |
| Boolean Chain (beam size=128) | 5.97% | 5.94% | 17.15 | 6.29 | 8.48% | 0.539 |
| AIGER | 8.35% | 7.77% | 17.19 | 6.34 | 8.46% | 0.025 |
| AIGER | 4.62% | 4.37% | 17.12 | 6.26 | 8.65% | 0.149 |
| AIGER | 4.23% | 3.98% | 17.12 | 6.25 | 8.59% | 0.906 |
| Circuit Transformer w/o Masking | 6.56% | 6.54% | 17.13 | 6.20 | 8.69% | 0.012 |
| Circuit Transformer w/o TPE | 6.63% | 0.00% | 17.33 | 6.46 | 7.46% | 0.019 |
| Circuit Transformer | 4.76% | 0.00% | 17.17 | 6.29 | 8.78% | 0.018 |
| Circuit Transformer ($K$=10) | 2.83% | 0.00% | 16.92 | 6.17 | 17.42% | 0.214 |
| Circuit Transformer ($K$=100) | **2.63%** | 0.00% | **16.73** | 6.07 | **26.68%** | 2.168 |
| Resyn2 (ground truth for training) | 0.00% | 0.00% | 16.83 | 5.57 | 0.00% | 0.008 |

| Reason to be Unsuccessful | Encoded Input Circuit Example | Encoded Output Circuit Example | Note |
|---|---|---|---|
| Equivalence constraint violation | 18 9 13 — 20 18 6 — 22 11 4 — 24 22 15 — 26 20 24 — 28 9 11 — 30 5 15 — 32 28 30 — 34 4 14 — 36 35 31 — 38 37 7 — 40 38 2 — 42 33 41 — 44 43 12 — 46 27 45 — 48 16 8 — 50 48 12 — 52 50 6 — 54 52 10 — 56 54 4 — 58 56 15 — 60 58 3 — [EOS] | 18 13 6 — 20 15 4 — 22 18 20 — 24 9 11 — 26 22 24 — 28 21 12 — 30 14 5 — 32 28 31 — 34 2 7 — 36 25 35 — 38 32 37 — 40 27 39 — 42 6 10 — 44 42 20 — 46 3 8 — 48 16 12 — 50 46 48 — 52 44 50 — [EOS] | No node is equivalent to the first output of the input circuit. |
| Not in valid AIGER format | 18 15 17 — 20 18 2 — 22 12 9 — 24 22 10 — 26 20 24 — 28 14 16 — 30 28 3 — 32 12 8 — 34 32 10 — 36 30 34 — 38 13 9 — 40 38 11 — 42 40 18 — 44 37 43 — 46 45 7 — 48 27 47 — 50 49 4 — 52 3 11 — 54 52 15 — 56 17 7 — 58 57 3 — 60 59 5 — 62 61 15 — 64 2 5 — 66 65 11 — 68 63 67 — 70 69 13 — 72 55 71 — 74 12 3 — 76 74 15 — 78 13 4 — 80 77 79 — 82 14 3 — 84 13 11 — 86 84 15 — 88 83 87 —, 90 89 7 — 92 80 91 — 94 93 17 — 96 72 95 — [EOS] | 18 9 2 — 20 10 12 — 22 18 20 — 24 15 17 — 26 22 24 — 28 9 13 —, 30 28 11 — 32 30 24 — 34 8 16 — 36 14 3 — 38 34 36 — 40 38 20 —, 42 33 41 — 44 43 7 — 46 27 45 — 48 47 4 — 50 15 3 — 52 50 11 —, 54 5 2 — 56 55 13 — 58 7 17 — 60 59 3 — 62 61 5 — 64 63 15 —, 66 65 10 — 68 56 67 — 70 53 69 — 72 50 12 — 74 4 13 — 76 73 75 —, 78 15 13 — 80 78 11 — 82 81 37 — 84 83 7 — **86 76 85 17** — 88 76 87 — [EOS] | 4 elements rather than 3 in the second last line of the output circuit (marked in bold type). |
| Exceeding maximal sequence length | 18 10 13 — 20 12 14 — 22 19 21 — 24 23 6 — 26 13 14 — 28 12 15 — 30 29 11 — 32 27 31 — 34 33 7 — 36 25 35 — 38 37 8 — 40 38 16 — 42 9 12 — 44 42 17 — 46 8 13 — 48 6 11 — 50 46 48 — 52 45 51 — 54 53 15 — 56 41 55 — 58 8 15 — 60 9 14 — 62 59 61 — 64 63 3 — 66 64 12 — 68 66 17 — 70 8 12 — 72 11 14 — 74 70 72 — 76 60 11 — 78 8 10 — 80 77 79 — 82 81 13 — 84 82 16 — 86 75 85 — 88 87 2 — 90 69 89 — 92 90 5 — 94 93 6 — [EOS] | 18 8 16 — 20 13 11 — 22 21 6 — 24 12 15 — 26 22 25 — 28 21 15 — 30 29 7 — 32 12 10 — 34 30 33 — 36 27 35 — 38 18 37 — 40 8 6 — 42 40 20 — 44 12 17 — 46 44 9 — 48 43 47 — 50 49 15 — 52 39 51 — 54 44 3 — 56 8 15 — 58 9 14 — 60 57 59 — 62 54 61 — 64 63 5 — 66 12 8 — 68 14 11 — 70 66 68 — 72 13 16 — 74 9 10 — 76 75 2 — 78 59 11 — 80 72 79 — 82 76 81 — 84 74 83 2 — 86 68 — 88 85 — 90 89 — 88 85 — 86 68 — 90 89 5 — 90 89 2 — 90 68 8 — 90 89 5 — 90 89 89 89 89 89 89 89 89 89 89 89 89 89 89 89 89 89 89 89 89 89 89 89 89 89 89 89 89 89 89 89 89 — 90 | No [EOS] appears in the first 200 tokens. |

Table 5: Example of unsuccessful cases for the baseline Transformer model trained on the AIGER format, taken from the IWLS FFWs dataset. "—" denotes the new line token. Index $2, 3, \ldots, 16, 17$ is reserved for $x_0, \overline{x_0}, \ldots, x_7, \overline{x_7}$. For the $i$-th AND node $a_i$ (start from 0), index $2(i + 9)$ denotes $a_i$ and index $2(i + 9) + 1$ denotes $\overline{a_i}$.

| Reason to be Unsuccessful | Encoded Input Circuit Example | Encoded Output Circuit Example | Note |
|---|---|---|---|
| Exceeding maximal sequence length | 18 19 18 18 2 15 18 6 16 18 18 5 11 18 9 13 19 19 18 19 18 18 9 13 5 18 18 3 14 18 7 17 19 18 18 18 19 19 18 4 12 9 19 19 19 5 12 19 4 13 8 6 16 2 15 19 19 19 18 5 13 8 19 19 19 5 12 19 4 13 9 19 19 18 2 15 18 7 17 19 18 3 14 18 6 16 10 18 18 18 19 3 9 14 4 12 1 [EOS] | 18 19 18 18 5 13 18 6 16 18 18 11 9 18 15 2 19 19 18 19 18 18 18 18 15 2 18 6 16 19 19 4 12 9 19 18 19 5 12 19 4 13 8 19 18 18 5 13 18 9 3 18 18 7 17 14 19 18 18 18 18 19 5 19 5 19 4 13 19 4 12 19 19 5 12 19 10 13 19 19 5 12 19 10 13 19 19 18 7 17 14 19 18 18 18 19 5 19 10 5 19 4 12 19 19 5 12 19 10 13 19 19 5 12 19 10 13 19 19 5 12 19 10 13 18 18 10 10 19 19 18 7 17 14 19 10 19 4 12 19 18 18 18 19 5 19 19 5 19 19 5 19 19 5 19 5 5 18 19 5 4 18 5 5 19 10 4 18 19 5 19 19 5 19 19 5 4 18 5 5 18 19 5 19 5 5 18 19 5 4 18 5 5 18 19 5 19 19 5 19 5 | While no equivalence constraint is violated, no [EOS] appears in the first 200 tokens. |

Table 6: Example of unsuccessful cases for the Circuit Transformer, taken from the IWLS FFWs dataset. Index $2, 3, \ldots, 16, 17$ denote $x_0, \overline{x_0}, \ldots, x_7, \overline{x_7}$. Index 18 and 19 denote $\wedge$ and $\overline{\wedge}$.

---

**Algorithm 5** Random generation of a $k$-input, $l$-output circuit

---

**Input:** Number of input $k$, number of output $l$, number of steps $T$.
**Output:** A randomly generated circuit with $k$ inputs and $l$ outputs.
  $C \leftarrow [I_0, I_1, \ldots, I_{k-1}]$
  **for** $i = 1, 2, \ldots, M_{\text{step}}$ **do**
    Create an AND node $s_i$
    Randomly sample two nodes $c_0, c_1 \in C$ without replacement
    Set the first input of $s_i$ as $c_0$ or $\overline{c_0}$ randomly
    Set the second input of $s_i$ as $c_1$ or $\overline{c_1}$ randomly
    Append $s_i$ to the end of $C$
  **end for**
  Return the circuit with $I_0, I_1, \ldots, I_{k-1}$ as primary inputs and $a_{T-l+1}, a_{M-l+2}, \ldots, a_T$ as primary outputs.

---

