# OpenReview forum: "Circuit Transformer: A Transformer That Preserves Logical Equivalence"
_ICLR.cc/2025/Conference — ICLR 2025 Poster_

### Official Review · Reviewer_yijD · 2024-10-29

**Soundness:** 3
**Presentation:** 2
**Contribution:** 3
**Rating:** 6
**Confidence:** 4

**Summary:**

The paper introduces a generative neural model (based on Transformers), which generates Boolean circuits equivalent to the model’s input. The primary contribution is a novel circuit representation coupled with a decoding mechanism that guarantees logical equivalence.

The sequential representation of circuits is based on a three-valued logic (true, false, unknown) and follows specific characteristics (Inheritability, Incrementality, Backtrack Elimination, Completeness). These characteristics ensure that 1) the model can build circuits incrementally while logical equivalence (under the tree-valued logic) is preserved in each step, and 2) no backtracking is necessary, which is on par with the generative properties of the Transformer.

A standard Transformer model is trained in a supervised manner to convert randomly constructed circuits to smaller but equivalent circuits (ground truth created by an existing tool). During inference, tokens that would invalidate the circuits (i.e., invalidate equivalence under three-valued logic) are masked and, hence, cannot be predicted. Consequently, every step of the generation produces a circuit equivalent to the input circuit under the three-valued logic, and every generated circuit that does not contain wild-card nodes (“unknown”) is equivalent to the input under boolean logic.

In addition to greedy decoding, the authors introduce a Monte-Carlo Tree Search (MCTS) to prefer circuits with fewer gates. The MCTS utilizes the next token probabilities of the Transformer and a reward function based on the number of gates in the circuit's DAG representation.

Experimentally, the authors’ approach outperforms Transformer models on other sequential and non-sequential representations of circuits. Using MCTS during inference also produces smaller circuits than the symbolical tool Resyn2 used for training data.

**Strengths:**

Representing circuits sequentially with cutoff properties is novel and integrates well with sequential generative methods such as Transformers. Formally guaranteeing the correctness of a generative solution is immensely important for circuit design. The authors show that an inherently sound generative process is beneficial compared to validating solutions past generation.

Integrating symbolic methods into the generative process not only gives formal guarantees on the result but also improves performance, which is an exciting finding and a strength of the paper.

The authors evaluate in-distribution on random circuits and out-of-distribution on circuits extracted from IWLS 2023 benchmarks, showing that their approach is robust.

**Weaknesses:**

## Related Work

I believe the related work (line 155ff, Schmitt et al. d’Ascoli et al.) is relevant because of the similarity in the domains and the usage of Transformers. While I agree that feasibility guarantees are important for tasks requiring correctness, it may be challenging to attribute the failure cases in related works solely to their absence, given the differences in task complexities. Both related works target very different tasks than this work. Because of the incomparability of the tasks and their complexity, it is unclear whether the failure cases are attributed to missing feasibility guarantees (as claimed by the authors) or the different tasks and complexities.

__Q1:__ Since the authors claim that the related work would benefit from feasibility guarantees, it would be interesting to see whether they think their approach also applies to the work of Schmitt et al. and d’Ascoli et al.

## Markov Decision Process and Monte Carlo Tree Search

The explanation of the generation using a Markov Decision Process (MDP) and Monte Carlo Tree Search (MCTS) should be expanded for clarity. Crucial details on how the trained Transformer model interacts with the Monte Carlo Tree Search are left in the Appendix (Algorithm 3). I recommend including MCTS in Section 3.5 and explaining how the Reward, the MCTS, and the Transformer model interact.

Additionally, algorithm 3 in the appendix could be more precise. E.g., When creating child nodes, the authors refer to $P(s_t|s_1, …, s_{t-1})$; however, $s_1$ to $s_{t-1}$ should be indexed by the MCTS nodes (line 878). Otherwise, P would be constant. Furthermore, the authors should clarify how “visited” and “total value” are updated (line 881). $s_a$ (line 870) has the same notation as a token, although it is a rational number.

## Experiments

It is noteworthy that an out-of-distribution real-world dataset has been evaluated. The experiments with different representations (Boolean Chain, AIGER, w/o TPE) show rigor in the evaluations. Providing details on the model’s runtime and scalability would add depth to the empirical results, particularly in comparison to Resyn2. These additions would provide a more comprehensive view of the method's practical implications, although they may not impact the current findings directly.

__Q2:__ The authors only report on average circuit sizes, but it would also be interesting to report the standard deviation. What percentage of circuits is smaller than the ground truth reference?

__Q3:__ How long does inference of the model take? If I understood correctly, the MCTS makes at least K independent calls to the model. Furthermore, how computationally expensive is the token masking in each step? Am I right that it is exponential in the number of inputs, as mentioned in Appendix 2? Generally, some time measurements of the approach would be interesting, especially in comparison with Resyn2.

__Q4:__ The approach can handle circuits with at most eight inputs and two outputs. Can the authors argue whether this is representative of the problem area? How many inputs/outputs does the IWLS benchmark (before extracting) have? __Q5__: Would the approach scale to larger circuits?

The following two questions did not affect my review but would be an interesting addition to the experiment section:

__Q6:__ It would be interesting to know how well the Circuit Transformer performs without next-token masking. This would singularize whether performance improvement is mainly based on the representation, the symbolic next-token masking, or the MCTS.

__Q7:__ The authors compare with other representations and apply beam search in those experiments. The authors acknowledge that this leads to a performance boost. Why is beam search not used in combination with the Circuit Transformer model?

## Presentation:

Although the paper is well-structured overall, several presentation adjustments could enhance readability, particularly in the methods section. Introducing symbols closer to their relevant subsections, providing an introductory paragraph for Section 3, and clarifying Figure 3 would help guide the reader. Such refinements would further support the clarity of the contributions. However, most of these issues are easy to fix and did not impact my assessment of the technical contribution and its soundness.
 - Most symbols are defined in the paper's first paragraph. Personally, I would use this space to motivate the paper rather than introduce formal definitions. I recommend moving symbol definitions closer to where they are used in the methods section so the reader does not have to search for those definitions.
 - An introduction before the first subsection of the method section would clarify the structure and guide the reader through the different subsections.
 - I’m not sure if the eight-queen-puzzle example as a vessel to explain cutoff properties is the right fit, as the reader expects circuits to be the domain of the paper. If the authors think introducing the eight-queen-puzzle is necessary, I would inform the reader before the subsection with a quick introduction that the puzzle is only an illustration to explain constrained sequence generation and that the sequential representation of circuits will follow in 3.2.
 - Figure 3 needs more explanation for easier understanding. That includes an intuitive explanation of $S_t$ and $s_t$. Furthermore, at this point, it is unclear to the reader why the unknowns are explored in this order, as it will be introduced later in 3.4. A note to this fact would help.
 - Section 3.3 starts with “in the last section”, but this is not the last section.
 - Important information (such as the loss function) is only mentioned in the caption of Figure 5. I recommend moving this to the main text.

## Summary

Overall, the authors present a novel and promising approach. Integrating feasibility guarantees with Transformer-based generative models represents a valuable contribution to the field. The experiment section is exhaustive regarding experiments on different representations and datasets. Comments/experiments about size and scaling would, however, strengthen the empirical results. Most of the issues in the presentation can be fixed by adding small explanations and some structure to guide the reader in the methods section. Section 3.5 should be updated, as crucial information is hidden in the appendix. The authors should be more accurate in the related work.

Weighing the idea, its relevance, and the contribution to the presentation, I am giving a weak acceptance, with the option to raise my rating if the authors address my questions and remarks and improve the presentation and experiments.

**Questions:**

See above.

---

> ### Author Response · Authors · 2024-11-22
>
> Thank you for your very constructive comment and your recognization of our work. The detailed response is as follows:
>
> ## Experiments
>
> We reran all the experiments to measure the time cost, and collect more statistics information including
>
> - Average inference time per circuit
> - Percentage of circuits that is smaller than the ground truth
> - Standard deviation of circuit size
> - The performance of Circuit Transformer without next-token masking
>
> The details are provided below and in Appendix A.8
>
> ### Random Circuits
>
> | Methods | Unsuccessful cases | Violation cases | Average circuit size | SD of  circuit size | % of circuits  strictly smaller  than Resyn2 | Average time  per circuit  (s) |
> |---|---|---|---|---|---|---|
> | Boolean Chain | 5.07% | 5.07% | 15.25  | 4.52  | 15.49% | 0.010  |
> | Boolean Chain (beam size=16) | 2.16% | 2.16% | 14.89  | 3.86  | 16.42% | 0.048  |
> | Boolean Chain (beam size=128) | 1.91% | 1.91% | 14.87  | 3.82  | 16.57% | 0.369  |
> | AIGER | 4.32% | 4.32% | 15.14  | 6.34  | 16.22% | 0.018  |
> | AIGER | 1.85% | 1.85% | 14.87  | 3.78  | 16.06% | 0.097  |
> | AIGER | 1.71% | 1.71% | 14.86  | 3.73  | 16.25% | 0.734  |
> | Circuit Transformer w/o Masking | 2.59% | 2.59% | 14.95  | 3.83  | 15.49% | 0.010  |
> | Circuit Transformer w/o TPE | 2.14% | 0.00% | 15.02  | 3.83  | 16.44% | 0.018  |
> | Circuit Transformer | 1.14% | 0.00% | 14.79  | 3.48  | 16.84% | 0.018  |
> | Circuit Transformer ($K$=10) | 0.20% | 0.00% | 14.02  | 2.79  | 36.15% | 0.210  |
> | Circuit Transformer ($K$=100) | **0.17%** | 0.00% | **13.73**  | 2.62  | **46.77%** | 2.090  |
> | Resyn2 (ground truth for training) | 0.00% | 0.00% | 14.56  | 2.98  | 0.00% | 0.009  |
>
> ### IWLS Fanout-Free Windows
>
> | Methods | Unsuccessful cases | Violation cases | Average circuit size | SD of  circuit size | % of circuits  strictly smaller  than Resyn2 | Average time  per circuit  (s) |
> |---|---|---|---|---|---|---|
> | Boolean Chain | 11.36% | 11.27% | 17.24  | 6.40  | 7.71% | 0.015  |
> | Boolean Chain (beam size=16) | 6.34% | 6.29% | 17.15  | 6.29  | 8.42% | 0.079  |
> | Boolean Chain (beam size=128) | 5.97% | 5.94% | 17.15  | 6.29  | 8.48% | 0.539  |
> | AIGER | 8.35% | 7.77% | 17.19  | 6.34  | 8.46% | 0.025  |
> | AIGER | 4.62% | 4.37% | 17.12  | 6.26  | 8.65% | 0.149  |
> | AIGER | 4.23% | 3.98% | 17.12  | 6.25  | 8.59% | 0.906  |
> | Circuit Transformer w/o Masking | 6.56% | 6.54% | 17.13  | 6.20  | 8.69% | 0.012  |
> | Circuit Transformer w/o TPE | 6.63% | 0.00% | 17.33  | 6.46  | 7.46% | 0.019  |
> | Circuit Transformer | 4.76% | 0.00% | 17.17  | 6.29  | 8.78% | 0.018  |
> | Circuit Transformer ($K$=10) | 2.83% | 0.00% | 16.92  | 6.17  | 17.42% | 0.214  |
> | Circuit Transformer ($K$=100) | **2.63%** | 0.00% | **16.73**  | 6.07  | **26.68%** | 2.168  |
> | Resyn2 (ground truth for training) | 0.00% | 0.00% | 16.83  | 5.57  | 0.00% | 0.008  |
>
> For the time cost of masking, they take 6% of the time cost for all the settings regarding Circuit Transformer. We also added a detailed analysis of the time complexity of masking in Appendix A.3 and Algorithm 3.
>
> For the selection of 8-inputs, 2-outputs as the circuit size, the motivation comes from the sub-circuit size selection in real-world circuit optimization. For real circuits with millions of nodes, given the NP-hardness of equivalence checking, it is unlikely that there exists any practical end-to-end method to optimize them as a whole with equivalence preserved. Generally speaking, decomposition is a must in practice. Cases in the IWLS benchmark have up to 16 inputs and various output setting from 1 to 103. To optimize circuits with such a size, a mostly used approach is to iteratively select small sub-circuits, and replace them with more compact ones (termed "rewriting"). The sub-circuit is typically of a single output, and with 4, 5 or 6 inputs [1, 2, 3]. Therefore,
>
> - We choose 8 inputs to be significantly larger than the 6-input setting, the largest sub-circuit setting to our knowledge
> - We choose 2-output to show that we can tackle multiple-outout circuits
>
> So back to the question, while this work will not directly scale to large circuits, it can serve as a "sub-circuit solver" that lies in the core of industrial circuit optimizers. The integration of Circuit Transformer into industrial circuit optimizers is another non-trivial research topic, which has been addressed in one of our next work.
>
> [1] Alan Mishchenko, et al., "DAG-aware AIG rewriting a fresh look at combinational logic synthesis", DAC'06
>
> [2] N. Li and E. Dubrova, "AIG rewriting using 5-input cuts", ICCD 2011
>
> [3] H. Riener, et al., "On-the-fly and DAG-aware: Rewriting Boolean Networks with Exact Synthesis", DATE 2019

---

> > ### Author Response · Authors · 2024-11-22
> >
> > We also reported the performance of Circuit Transformer without masking in the above tables. Note that the decrease of time cost is larger than 6%, due to that masking mechanism tends to "force" the model to prefer longer but feasible circuits (which takes more time to inference), rather than shorter but invalid ones.
> >
> > For why beam search is not used in combination: beam search should also work on Circuit Transformer. The main reason is that we believe MCTS is a better choice. Beam search aims to find the most probable sequence, which helps the model better "imitate" the supervised signal (i.e., Resyn2), but cannot surpass the supervised signal on optimality (i.e., circuit size). Nonetheless, it is very possible for us to implement beam search for Circuit Transformer in the future, especially for some scenarios that time cost is not so sensitive (e.g., IWLS contest benchmark).
> >
> > ## Presentation
> >
> > Thank you for your very detailed suggestions on presentation. We have updated the paper as follows:
> >
> > - We updated the section regarding Markov decision process and Monte-Carlo tree search. Algorithm 3 is heavily refined for clarity and preciseness, and moved to Section 4.5 (now it is Algorithm 2). More explanation is added to clarify how the reward, MCTS and Circuit Transformer interact.
> > - We updated the introduction section. Formal definitions are moved to related sections (Section 3 and Section 4.5), and the introduction is refined to focus more on motivating the paper.
> > - We updated the related work for preciseness. Additionally, we think our proposed approach can be applied to the work of Schmitt et al. and d'Ascoli et al with some adaptation. For the former, the cutoff properties should be refined to accomodate LTL. For the latter, the domain $D$ should be extended to accomodate more Boolean operators. The constraints should be relaxed to be on K given points rather than on all possible $2^N$ input values for symbolic regression.
> > - We added an introduction before the first subsection of the method section to guide the reader.
> > - Some more modifications including adding notes to the caption of Figure 3, the introduction of loss function, and fixing writing errors.
> >
> > Thank you again for your detailed review. Pleas feel free to ask if you have any further questions or require additional information.

---

> ### Author Response · Authors · 2024-11-26
>
> Dear Reviewer yijD,
>
> Thank you for your thoughtful feedback on our submission. We hope that our detailed responses have addressed your concerns, including the request for additional experimental details (such as inference time, standard deviation of circuit size, and scalability discussions) as well as improvements to the presentation (such as expanding the MDP and MCTS section, and refining the introduction, related work, and methods sections). If there are any remaining issues or points requiring further clarification, please do not hesitate to let us know.
>
> Best regards,
>
> The Authors

---

> ### Author Response · Authors · 2024-11-30
>
> Dear Reviewer yijD,
>
> Thank you for your thoughtful and constructive comments on our submission. As the discussion deadline approaches, we kindly seek your feedback on whether our rebuttal has sufficiently addressed your concerns. Please feel free to let us know if there are any remaining issues or points that require further clarification.
>
> Best regards,
>
> The Authors

---

### Official Review · Reviewer_w2Zd · 2024-11-04

**Soundness:** 3
**Presentation:** 3
**Contribution:** 3
**Rating:** 6
**Confidence:** 3

**Summary:**

This paper presents a generative model to produce circuits while ensuring logical equivalence with the given Boolean function. The key innovation is the "Circuit Transformer", which utilizes a novel decoding mechanism based on cutoff properties that enable token generation in a way that prevents logical errors. This ensures each generated circuit is logically equivalent to its Boolean function. The Circuit Transformer was tested against existing tasks and achieved good performance in terms of both success rate and circuit size.

**Strengths:**

- A novel transformer-based method to generate a circuit while preserving the semantics of the original Boolean function.
- Experiments show good performance compared with other methods.

**Weaknesses:**

- The motivation and intuition behind the proposed method are not immediately clear. Including a concrete illustrative example or practical application would significantly aid in understanding the approach.
- The authors provide an example illustrating how a feasible circuit can be constructed using the proposed method, but the process for selecting each token $s_i$ is not explicitly explained in the example. This leaves ambiguity about whether the selection is deterministic or nondeterministic, as I could not locate specific instructions for this process in the paper. More details would be beneficial to help understand the cutoff-property-based method.
- The set $S_i$ is central to ensure the equivalence between Boolean functions and generated circuits, as it necessitates checking that $F(\cdot)$ holds (essentially a semi-equivalence SAT check) after each token selection (line 3 in Alg.1). Given the top-down mechanism, I guess it will become increasingly non-trivial? It is also unclear how this "semi-equivalence checking" is guaranteed within the Transformer model, particularly in terms of maintaining the inference efficiency. Additionally, Table 2 does not include training and inference time, which would provide valuable insight into efficiency.
- Given that the benchmark circuits involve 8-input and 2-output nodes, it seems feasible to represent such boolean functions using a more compact form, such as Binary Decision Diagram (BDD). Existing methods for converting BDDs into circuits while preserving Boolean function semantics could serve as a viable alternative approach. Why were these methods not considered in your comparative experiments?

**Questions:**

Please address the weakness raised in **Weaknesses**.

---

> ### Author Response · Authors · 2024-11-22
>
> Thank you for your constructive comment and your recognition of novelty. The detailed response is as follows:
>
> 1. We have updated the introduction section, moving formal definitions to other sections and highlighting the motivation. We hope this will be of help to readability.
> 2. In Circuit Transformer, $s_t$ is selected from $S_t$ by selecting from the masked probability distribution $P_\text{CT}(\cdot | s_1, \dots, s_{t-1} ; f)$. The selection process can be both deterministic (greedily select the one with the largest probability) or nondeterministic (sample from the distribution). In the experimental section, we choose to be deterministic for the sake of reproductivity of the result. We added explansions in both the caption of Figure 3 and Section 5 (line 472) to make it clear to the readers.
> 3. We added a detailed introduction for the computation of $S_t$ in Appendix A.3 and Algorithm 3, which is of time complexity $O(N \cdot 2^N \cdot d)$, in which $N$ is the number of inputs and $d$ is the depth of the wildcard node at time step $t$. We implemented a caching mechanism to ensure that the time complexity depends on the depth of the current node rather than the total number of nodes, as the depth is typically significantly smaller. More specifically, the computational process consists of $d$ times of element-wise Boolean computation between two $N \times 2^N$ matrices, which can be processed efficiently via vectorization libraries when $N$ is not large. For training time, all the models are trained for 75 hours on a single NVIDIA GeForce RTX 4090 graphic card, which we added in line 501. We also provided more detailed experimental results in Appendix A.8. (Table 3 and 4) including the inference time cost. The time cost of masking layer contributes to 6% of the total time cost. Experimental results show that the time cost of Circuit Transformer is close to other Transformer models, and is comparable to resyn2 under the current model size setting of 88 millions. For methods combined with heuristics search, the time cost is proportional to the number of search rounds or beam size.
> 4. In the experimental section, we mainly focus on neural generative approaches that convert Boolean functions to circuits, and demonstrate that our proposed neural generative method can preserve the equivalence as we claimed, while other neural approaches cannot. Existing symbolic methods are mainly served as training signals / ground truth, rather than baselines to be directly compared. However, BDD is indeed a worthwhile representation for its compactness, which should be further investigated in the future. We added it in line 529 of the paper.
>
> Thank you again for your review. Pleas feel free to ask if you have any further questions or require additional information.

---

> ### Author Response · Authors · 2024-11-26
>
> Dear Reviewer w2Zd,
>
> Thank you for your thoughtful feedback on our submission. We hope our detailed responses have effectively addressed your concerns, including but not limited to whether the selection of $s_t$ is deterministic, the computational process & time complexity of $S_t$, and the running time statistics. If there are any remaining issues or points requiring further clarification, please do not hesitate to let us know.
>
> Best regards,
>
> The Authors

---

> ### Author Response · Authors · 2024-11-30
>
> Dear Reviewer w2Zd,
>
> Thank you for your thoughtful and constructive comments on our submission. As the discussion deadline approaches, we kindly seek your feedback on whether our rebuttal has sufficiently addressed your concerns. Please feel free to let us know if there are any remaining issues or points that require further clarification.
>
> Best regards,
>
> The Authors

---

### Official Review · Reviewer_Gw9o · 2024-11-05

**Soundness:** 2
**Presentation:** 2
**Contribution:** 3
**Rating:** 8
**Confidence:** 4

**Summary:**

Summary:
This paper introduces an approach called "Circuit Transformer" for generating logically equivalent circuits while preserving strict equivalence constraints. Existing neural approaches often make occasional errors, violating the equivalence.

**Strengths:**

a.	A decoding mechanism that builds circuits step-by-step using tokens with "cutoff properties" to block invalid tokens.
b.	A "Circuit Transformer" model incorporates this mechanism as a masking layer.
c.	Formulation of equivalence-preserving circuit optimization as a Markov decision process

**Weaknesses:**

1.  It is not clear why we need circuit transformer. What does this approach buy compared to existing commercial tools for performing logic synthesis from truth tables? There are methods like the Quine-McCluskey method and its many optimized variants built into commercial logic synthesis tools that can handle Boolean functions of a much larger scale than what this paper proposes. Without a comparative analysis with logic synthesis, this paper's motivation is unclear.
2.	Scalability: While the paper demonstrates effectiveness on 8-input, 2-output circuits, it's unclear how well the approach scales to much larger circuits with hundreds or thousands of inputs/outputs. In any modern digital design, there are likely to be billions of logic gates. What is the point of illustrating on an 8-input 2-output circuit?
3.	Computational: The paper does not provide a detailed analysis of the computational complexity of the proposed approach, especially for larger circuits. This information would be valuable for assessing practical applicability.

**Questions:**

Please address the weaknesses highlighted.

---

> ### Author Response · Authors · 2024-11-22
>
> Thank you for your constructive comment from the prespective of applicability. The detailed response is as follows:
>
> For why we need circuit transformer, we would like a chance to clarify our main motivation. This work mainly contributes to the AI community by integrating feasibility guarantees with Transformer-based generative models. Generative models like ChatGPT are often criticized for their lack of preciseness, while this paper shows that formal guarantee of preciseness is achievable on certain tasks with cutoff properties. We acknowledge that mature, powerful logic synthesis systems do exist in the EDA industry, and our proposed Circuit Transformer, _on its own_, is not a direct competitor of such systems. However, as a submission to a representation learning conference, we believe that representing circuits sequentially with cutoff properties is novel, and a formal preciseness guarantee of generative neural models is of interest to the ICLR community. To better motivate the paper, we updated the introduction section, moving formal definitions to other sections to highlight the contribution.
>
> Nonetheless, we would like to discuss how Circuit Transformer contributes to the EDA industry, as this may help explain why this paper illustrates on 8-input 2-output circuits. Generally speaking, for modern digital design with billions of gates, it is unlikely for _any_ synthesis tools (not only Circuit Transformer but also other commertial tools) to optimize them _as a whole_. In practice, decomposition is a must. A mostly applied decomposition approach is to iteratively select small sub-circuits, and replace them with more compact ones (termed "rewriting"). The sub-circuit is typically of a single output, and with 4, 5 or 6 inputs [1, 2, 3]. Therefore,
>
> - We choose 8-input to be significantly larger than the 6-input setting, the largest sub-circuit setting to our knowledge
> - We choose 2-output to show that we can tackle multiple-outout circuits
>
> While this work will not directly scale to large circuits (mainly due to the NP-hardness of equivalence checking in the masking layer), it can serve as a "sub-circuit optimizer" that lies in the core of industrial circuit optimizers. Instead of traditional optimizers that pre-compute a database of all possible occurence of sub-circuits (for efficiency), or optimize with exact SAT solvers every time from scratch (for flexibility), Circuit Transformer lies in between. It is a pre-computation technique that is trained on millions of circuits in advance and runs efficiently, while it also supports important flexibilities in sub-circuit optimizers such as don't cares [3] and logic sharing [1]. The integration of generative neural networks into industrial circuit optimizers is another non-trivial research topic, which has been addressed in one of our next work with Circuit Transformer and Monte-Carlo tree search, and evaluated on the IWLS benchmark for effectiveness.
>
> For computational analysis:
> - Theoretically, we proposed a cache-based approach to compute the masking layer (i.e., $S_t$) in a time complexity of $O(N \cdot 2^N \cdot d)$, in which $N$ is the number of inputs and $d$ is the depth of the wildcard node at time step $t$. More specifically, it is $d$ times of element-wise Boolean computation between $N \times 2^N$ matrices, which can be processed efficiently via vectorization libraries when $N$ is not large. The details are provided in Appendix A.3 and Algorithm 3.
> - Experimentally, we provided more detailed results including the running time for all the methods, which is shown in Appendix A.8. The time cost of masking layer contributes to 6% of the total time cost. Experimental results show that the time cost of Circuit Transformer is close to other Transformer models, and is comparable to resyn2 under the current model size setting of 88 millions. For methods combined with heuristics search, the time cost is proportional to the number of search rounds or beam size.
>
> While optimizing larger circuits is not the main target of this paper, it has been fully addressed in our aforementioned sequel work.
>
> Thank you again for your review. Pleas feel free to ask if you have any further questions or require additional information.
>
> [1] Alan Mishchenko, et al., "DAG-aware AIG rewriting a fresh look at combinational logic synthesis", DAC'06
>
> [2] N. Li and E. Dubrova, "AIG rewriting using 5-input cuts", ICCD 2011
>
> [3] H. Riener, et al., "On-the-fly and DAG-aware: Rewriting Boolean Networks with Exact Synthesis", DATE 2019

---

> ### Author Response · Authors · 2024-11-26
>
> Dear Reviewer Gw9o,
>
> Thank you for your thoughtful feedback on our submission. We hope our detailed responses have effectively addressed your concerns about motivation (why we need circuit transformer) and applicability (discussion regarding scalability, as well as computational analysis). If there are any remaining issues or points requiring further clarification, please do not hesitate to let us know.
>
> Best regards,
>
> The Authors

---

> > ### Comment · Reviewer_Gw9o · 2024-11-27
> > **Reservations remain**
> >
> > I appreciate the thorough response that the authors have written and the proposed approach does have some merits.
> >
> > However, I remain unconvinced about the need for a circuit transformer. In the rebuttal, the authors suggest that it could be used for sub-circuit optimization by performing decomposition. Automatically performing decomposition seems like the hardest challenge here. Once the decomposition is performed, existing tools in the EDA industry would give guaranteed, precise solutions without any worries about errors introduced by Gen AI. It is not clear why a designer would use a new (and potentially unreliable) tool with questionable scalability when existing tools do the job (for sub-circuit optimization)?
> >
> > Cutoff properties seem useful: perhaps you can use them to integrate other kinds of logic-based reasoning in Gen AI?

---

> > > ### Author Response · Authors · 2024-11-28
> > >
> > > Thank you very much for your further comments.
> > >
> > > While the significance of Circuit Transformer is more obvious for the AI community via introducing formal feasibility guarantee to Gen AI, we acknowledge that more elaboration may be needed for its benefit to the EDA industry. Particularly, we need to know how current EDA tools optimize sub-circuits, what their downsides are, and how Circuit Transformer addresses these issues. The explanation may be a bit long, including the introduction to key concepts like **pre-computation**, **don't cares** and **on-the-fly exact synthesis**. We thank you in advance for your patience.
> > >
> > > As we have introduced before, decomposition is a must for large circuit optimization. When the decomposition reaches a certain stage, sub-circuits should not be further decomposed anymore. Instead, they should be optimized _as a whole_, which we call "sub-circuit optimization". A common sense under such a setting is that, larger considered sub-circuits has more potential in creating better optimized circuits [4]. However, while existing EDA tools can optimize large circuits, a counterintuitive fact is that, the size of sub-circuits that they can optimize _as a whole_ is typically very small. A widely adopted approach is **pre-computation**, in which all possible occurence of sub-circuits are pre-optimized and stored in a database. For example, 4-input, 1-output circuits can represent $2^{2^4} = 65536$ different Boolean functions. Then, for all possible Boolean functions, we compute their size-optimal circuit implementations, and store them in a database. When we have a large circuit to optimize, we decompose it to numerous 4-input, 1-output sub-circuits, which can be optimized by simply query the database and get the equivalent (i.e., representing the same Boolean function) size-optimal circuits [1]. To our knowledge, the largest setting for pre-computation is 5-input, 1-output [2], which can represent $2^{2^5} \approx 4.29 \cdot 10^9$ Boolean functions. It can be further reduced to 616,126 "NPN equivalence classes", and shrinked by only storing frequently used ones.
> > >
> > > While very efficient, pre-computation has significant disadvantages. Apart from its tiny scale, it is also inflexible. For a sub-circuit to be optimized, it can only fetch the size-optimal one that _exactly_ match the original sub-circuit. However, once the functionality of the whole circuit is precisely maintained, a sub-circuit can be replaced by a _non_-equivalent one. I.e., for the $2^N$ equivalence constraints in Eq (2) of our paper, only part of them need to be obeyed (termed "care set") and the others are not (termed "**don't cares**" [5]). To exploit such a flexibility, recent works [3, 6] optimize a sub-circuit with exact SAT solvers every time from scratch, which is named **on-the-fly exact synthesis**. It is also a bit more scalable than pre-computation, for which the maximal sub-circuit size is 6-input and 1-output to our knowledge, but with significant time cost.
> > >
> > > Back to our proposed Circuit Transformer. From the perspective of sub-circuit optimization, it is a **pre-computation** technique. The Circuit Transformer, which is pre-trained on millions of optimized circuits in advance, can be regarded as a "soft database" of optimized circuits that can be queried efficiently. However, its capacity is much larger, allowing a size setting of 8-input, 2-output, which can specify $(2^{2^8})^2 = 1.34\cdot 10^{154}$ different Boolean functions. This is similar to the Q network of deep reinforcement learning compared with traditional tabular-based approaches. In the meantime, Circuit Transformer supports **don't cares**, with a small modification of Eq (5) of our paper from $\forall x \in \\{0, 1\\}^N$ to $\forall x \in \\{\text{care set}\\}$, providing more flexibilities in optimizing sub-circuits. Moreover, the optimality of Circuit Transformer can be further boosted by MCTS once the time budget is sufficient. One of our sequel work further demonstrates the great effectiveness of Circuit Transformer over traditional sub-circuit optimization approaches on the IWLS 2023 benchmark.
> > >
> > > For the cutoff properties, while this is the first paper intergrating such properties in Gen AI, we are also considering the potential of such methodology in other logic reasoning tasks. An important task is 0-1 Mixed Integer Linear Programs, in which a set of Boolean variables need to be optimized w.r.t certain objective, while some linear constraints must be satisfied.
> > >
> > > We hope the above explanation can address your concern. Please do not hesitate to let us know if you have further questions.
> > >
> > > [4] Xuliang Zhu, et al. A Database Dependent Framework for K-Input Maximum Fanout-Free Window Rewriting, DAC'23
> > >
> > > [5] Alan Mishchenko, et al., Scalable don't-care-based logic optimization and resynthesis, ACM Trans. Reconfigurable Technol. Syst.
> > >
> > > [6] H. Riener, et al., Exact DAG-aware rewriting, DATE'20

---

> > > > ### Comment · Reviewer_Gw9o · 2024-12-02
> > > > **Thanks for the explanation**
> > > >
> > > > Thanks for the detailed explanation, this makes things a bit clearer. I would highly suggest adding experiments that do apples-to-apples comparisons of existing EDA techniques with your approach, to show the clear benefit of your approach. Without such experiments, questions like the ones I asked may arise. I am happy to raise my score based on the current explanation.

---

> > > ### Author Response · Authors · 2024-12-02
> > >
> > > Dear Reviewer Gw9o,
> > >
> > > Thank you for your further comments on our submission about the need for a Circuit Transformer. As the discussion deadline approaches, we kindly seek your feedback on whether our explanation above has sufficiently addressed your concerns. Please feel free to let us know if there are any remaining issues or points that require further clarification.
> > >
> > > Best regards,
> > >
> > > The Authors

---

### Author Response · Authors · 2024-11-25
**Global Response**

Dear reviewers and AC,

We really appreciate all the three reviewers (Gw9o, w2Zd and yijD) for their constructive comments. In this paper, we proposed a novel representation of circuits with "cutoff properties", allowing Transformer-based generative models to generate equivalent circuits with **formal guarantee**.

In the review process, while we are encouraged by the recognition of novelty, we are also aware of a lack of detailed computational efficiency analysis in this paper. To fully address reviewers' concern, we made the following key changes to improve the paper:
1. Theoretically, we extend the discussion of Characteristic 4.2 in Appendix A.2, **elaborating how the time complexity of $O(N \cdot 2^N \cdot d)$ is achieved** for the masking layer of Circuit Transformer ($N$ is the number of inputs, $d$ is the depth of the wildcard node to replace at step $t$). A thorough description of $S_t$'s computation process is is included in **Appendix A.3** and **Algorithm 3**, showing how a cache mechanism plays a critical role for efficiency.
2. Experimentally, we reran all the experiments to measure the time cost, which can be found in **Appendix A.8**. Importantly, we measured that the time cost of the masking layer contributes to **6%** of the total time cost, which is not a computational bottleneck of the proposed model. Experimental results show that Circuit Transformer is efficient, whose time cost is not only close to other Transformer models, but also comparable to resyn2 under the current model size setting of 88 million parameters.

Some reviewers also point out that our scale of experiments (8-input, 2-output circuits) seems small compared to real-world circuits. While we have already pointed out that 8-input, 2-output circuits can specify $(2^{2^8})^2 = 1.34\cdot 10^{154}$ different Boolean functions (there are $10^{78} \sim 10^{82}$ atoms in the observable universe), we would like to make a further clarification from the perspective of digital design:
1. Generally speaking, no existing methods can optimize real-world circuits **as a whole**. There is no magic in industrial circuit optimizers to handle large circuits -- they need to **decompose a circuit to numerous small sub-circuits that can be directly optimized**, which are typically of a single output, and with 4, 5 or 6 inputs. We chose an 8-input, 2-output setting to be significantly larger than these typical settings, which can be served as a more capable "sub-circuit optimizer".
2. The integration of generative neural networks into industrial circuit optimizers is highly non-trivial, requiring specific digital design methods to be proposed to make it efficient and effective. This is why we do not include the method/result of real-world circuits in this paper, but choose to elaborate them in a seperate full paper to digital design venues.

We have also improved the representation of the paper according to the suggestions from the reviewers. We sincerely thank the reviewers for their valuable feedback on strengthening our work and remain open to further suggestions.

Thank you very much,

Authors.

---

### Meta-Review · Area_Chair_1vf4 · 2024-12-24

**Metareview:**

This paper introduces “circuit” transformer which generate logical circuits. Furthermore it can ensure the generated circuits’ equivalence with the given boolean function. The main novelty lies on its proposes decoding mechanism based on a cutoff property that prevents logical error. The technical contribution is solid, which is shared across all reviewers. The integration of symbolic methods into the generation process not only gives formal guarantees but also improve performance. This is a big strength. Theory and experiments math each other and it works. Both in-distribution and out-of-distribution results are good and robust. However, this paper is a dense one, which means its presentation can be further improved. Some of the notations can be more consistent, and introduced easier in the manuscript. And another trick that could make this paper even better is that, readers could be overwhelmed by the technical details and in turn forget the big picture and the motivation of this paper. In some sense, the readers could find it difficult to have a feel of the method. An illustrative example, which makes discussion more concrete, would help a lot.

**Additional Comments On Reviewer Discussion:**

Before the rebuttal, reviewers all appreciate this paper's technical contribution. The main concern lies in its presentation clarity, details regarding the experiments,and explanation about the formal guarantee (theorems) in the paper. The rebuttal almost addresses each of them and convinces the reviewers.

---

### Decision · Program_Chairs · 2025-01-22

Accept (Poster)